# CONTRASTIVE LEARNERS ARE SEMANTIC LEARNERS

## ABSTRACT

In this work, we explore the definition of semantic equivalence to establish a connection between contrastive tasks and their downstream counterparts. Specifically, we investigate when a contrastive dataset can learn representations that encode formal semantic equivalence relations for a specific downstream task. In our analysis, we recover a surprising hypothesis resembling the distributional one—dubbed distributional alignment hypothesis. Under this assumption, we demonstrate that the optimal model for simple contrastive learning procedure must generate representations that encode formal semantic equivalence relations for the downstream task. Furthermore, we support the theory with a series of experiments designed to test the presented intuitions.

## 1 INTRODUCTION

**Overview.**     Contrastive Learning has emerged as a prominent self-supervised training technique, demonstrating success across diverse data modalities. These include, among others, images (Chen et al. (2020); He et al. (2020)), audio (van den Oord et al. (2018); Saeed et al. (2021)), video (Dave et al. (2022); Pan et al. (2021)), text (Fang et al. (2020); Xiong et al. (2020)), graphs (Thakoor et al. (2021); Ling et al. (2023)), and time series (Zheng et al. (2024)). Notable contrastive learning techniques encompass SimCLR (Chen et al. (2020)), MoCo (He et al. (2020)), BYOL (Grill et al. (2020)), SwAV (Caron et al. (2020)), MSimCLR (Korman (2021)), and Barlow Twins (Zbontar et al. (2021)).

**Problem Statement.**     The primary goal of these techniques is to generate high-quality embedding representations that can be effectively utilized for downstream tasks (potentially with limited data availability). Empirical evidence suggests that semantically similar objects tend to be encoded into proximate embedding representations during contrastive training (Chen et al. (2020); Jaiswal et al. (2020)). While the emergence of semantic structure in these embeddings is well-documented (Le-Khac et al. (2020); Kang et al. (2020); Poklukar et al. (2022)), to the best of our knowledge, a formal treatment of this process remains lacking in the current literature.

**Proposed Solution.**     This work aims to address the aforementioned gap by formalizing the concept of *semantic equivalence*, as proposed by Bertolotti & Cazzola (2024) for sequence modeling tasks. Briefly, two symbols are defined as semantically equivalent if they can be substituted without altering the outcome distribution of a certain task (see Definition 2.1). This property is fundamentally rooted in the notion of semantic equivalence in programming languages (Scott & Strachey (1971)), where two code snippets are considered semantically equivalent if their substitution does not affect the program's output regardless of the context (borrowing the denotational semantics for an instant, we would write $\forall \rho : [\![p_1]\!](\rho) = [\![p_2]\!](\rho)$).

**Research Questions.**     To summarize, our research aims to address the following questions:

- **RQ$_1$: Under what conditions is a contrastive task useful for a downstream task?**
- **RQ$_2$: How can we train embeddings that effectively encode semantical equivalence relations?**

**Findings.**     Our findings demonstrate that the SimCLR contrastive learning procedure (Chen et al. (2020)) inherently encodes semantically equivalent symbols in close proximity within the embedding space. Furthermore, we find a fundamental requirement for a contrastive task to be useful for the downstream counterpart. We dub this requirement *distributional alignment hypothesis*, for its similarity with the classical *distributional hypothesis* (Bertolotti & Cazzola (2024)). These re-

sults provide a theoretical foundation for the empirically observed semantic structure in contrastive learning embeddings.

## 2 BACKGROUND & NOTATION

Let us begin by introducing some recurring notations. From a classical supervised learning stand-point, one often uses a data distribution from which input-label pairs are sampled, $x, y \sim \mathcal{D}$. In this work, we split the *input* component into *symbol* and *context* components. Therefore, we will use a data distribution from which we can sample symbol-context-label triplets, $\sigma, \rho, y \sim \mathcal{D}$. In practice, a symbol may correspond to a word, while the context may represent the rest of the sentence. Similarly, with images, a symbol may represent an image patch, while the context may represent the rest of the image. We will also use $\Sigma$, P, and $\mathcal{Y}$ to denote the symbol, context, and label domains respectively. We will use the notation $p_{\mathcal{D}}$ (or simply $p$ when the underlying data distribution is evident) to denote the probability function for data sampled according to $\mathcal{D}$. For example, $p_{\mathcal{D}}(y|\sigma, \rho)$ represents the probability of label $y$ when conditioned on the symbol context pair $(\sigma, \rho)$ when $\sigma, \rho$ and $y$ are sampled according to $\mathcal{D}$.

### 2.1 SEMANTIC EQUIVALENCE

Intuitively, two symbols $u$ and $v$ are semantically equivalent if we can use them interchangeably without affecting the output distribution, or formally:

**Definition 2.1** (semantic equivalence (Bertolotti & Cazzola (2024))). Given $u, v \in \Sigma$, $u \stackrel{\circ}{=} v$ (u is semantically equivalent to v) iff

$$\forall \rho \in \mathrm{P}, y \in \mathcal{Y} \, . \, p(y|u, \rho) = p(y|v, \rho)$$

For example, consider a simple Masked Language Modeling (MLM) tasks (Devlin et al. (2019)), and, consider the sentence

$$\texttt{the <MASK> of water is half \underline{empty}/\underline{full}.}$$

Here, we could use both `empty` and `full` without affecting the outcome distribution of the mask token. Therefore, we can say that, in this context, `empty` and `full` are semantically equivalent. If this relation were to hold true for all contexts, then, we would say that `empty` and `full` are semantically equivalent in general.

In a different scenario, consider a cat-vs-dog image classifier where each symbol represents an image patch—similar to the setup used in Vision Transformers (Dosovitskiy et al. (2020)). In this context, two image patches, such as one depicting a clouded sky and another showing a clear sky, would be considered semantically equivalent since they do not influence the classification outcome.

A similar notion of semantics is also used by Chiang & Yogatama (2023).

Throughout this work, we will often use the term semantic with its formal meaning in mind rather than it liguistic meaning. Therefore, unless stated otherwise, when speaking of semantic relations or semantic learning, we refer to the formal definition of semantic equivalence (Definition 2.1).

### 2.2 SIMPLE CONTRASTIVE LEARNING

SimCLR (Chen et al. (2020)) is a straightforward contrastive learning procedure initially proposed for image data (see Figure 1a). The process begins with a batch of $N$ data points $\{x_i\}_{i=1}^{N}$ sampled from the data distribution $x_i \sim \mathcal{X}$, and a corresponding batch of $N$ augmentation pairs $\{(t_i, t_i')\}_{i=1}^{N}$ sampled from a transformation distribution $t_i, t_i' \sim \mathcal{T}$. The objective is to maximize agreement between different augmentations of the same data point ($t_i(x_i)$ and $t_i'(x_i)$) while maximizing disagreement between different data points ($t_i(x_i)$ and $t_j'(x_j)$ for $i \neq j$). This is achieved through the following loss function (InfoNCE (van den Oord et al. (2018))):

$$\mathcal{L}(D) = -\frac{1}{N} \sum_{i=1}^{N} \log \left( \frac{\exp(\mathrm{sim}(E(t_i(x_i)), E(t_i'(x_i))))}{\sum_j \exp(\mathrm{sim}(E(t_i(x_i)), E(t_j'(x_j))))} \right) \tag{1}$$

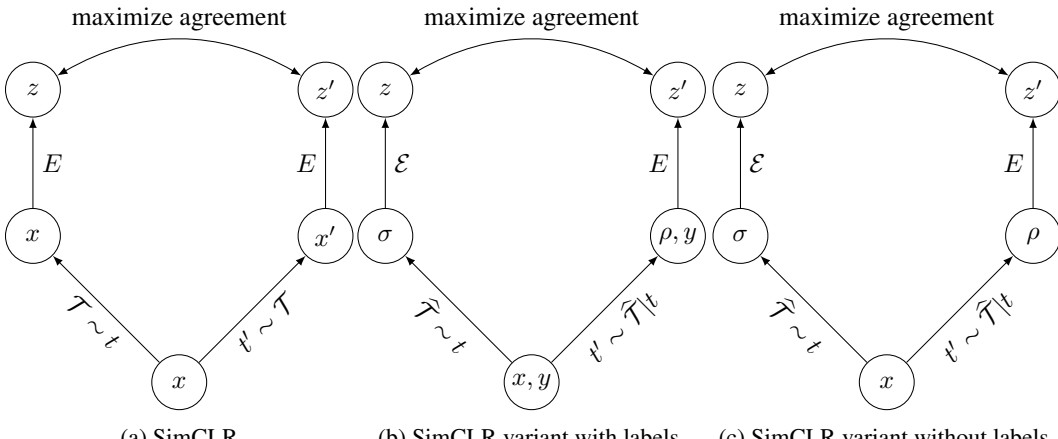

(a) SimCLR  (b) SimCLR variant with labels.  (c) SimCLR variant without labels.

Figure 1: Figure 1a illustrates the SimCLR architecture. Figure 1b shows an asymmetric Sim-CLR architecture, where two different encoder functions, $\mathcal{E}$ and $E$, are used in separate branches. Additionally, a different set of augmentations, $\widehat{\mathcal{T}}$, is applied, with the right branch's augmentation conditioned on the left branch. Figure 1c depicts the previous variants without the use of labels

Here, $E$ denotes an encoder network and $\mathrm{sim}$ represents a similarity function, typically scalar product or cosine similarity (we assume the former in this work). The i-th component in the summation is minimized when the similarity between $t_i(x_i)$ and $t'_i(x_i)$ is maximized, while the similarity between $t_i(x_i)$ and any other $t'_j(x_j)$ is minimized. The full loss $\mathcal{L}$ is obtained by averaging over the batch ($D$).

## 3 CONTRASTIVE LEARNERS (WITH LABELS) ARE SEMANTIC LEARNERS

Contrastive learning techniques typically assume that labels for downstream tasks are unavailable, as they are often costly to obtain. However, as we will show, if labels are provided to a simple variation of SimCLR, it is possible (under appropriate assumptions) to recover representations that encode semantic equivalence relations (as in Definition 2.1). This means that if two symbols are semantically equivalent, their representations will also be equivalent. In the next section, we will explore how to eliminate the need for labels in this framework.

The variant of SimCLR, depicted in Figure 1b, introduces two main differences. First, we employ two specific types of augmentation functions: $t_i : \mathcal{X} \to \Sigma$ and $t'_i : \mathcal{X} \to \mathrm{P}$. The former is responsible for extracting the $i$-th symbol from the input, while the latter returns the context of the $i$-th symbol. Here, the $i$-th symbol may represent the $i$-th image patch, and the $i$-th context may represent the remaining patches. Using SimCLR terminology, the first augmentation could be a *crop and resize*, resulting in the *symbol*, while the second could be a *Cutout* specifically removing the previous *symbol* to produce the *context*. Let us refer to the set of these augmentations with $\widehat{\mathcal{T}}$.

Secondly, we introduce an asymmetric architecture to SimCLR. Specifically, we denote the embedding function $\mathcal{E} : \Sigma \to \mathbb{R}^d$, which maps symbols to a $d$-dimensional representation. Meanwhile, let $E : \mathrm{P} \times \mathcal{Y} \to \mathbb{R}^d$ represent the function that maps context-label pairs to a $d$-dimensional representation. The parameters between the embedding function and the encoder function can be shared, partially shared, or separate.

Consider a batch $D = \{(\sigma_i, \rho_i, y_i)\}_{i=1...N}$ consisting of symbol-context-label triplets, where $\sigma_i = t(x)$ and $\rho_i = t'(x)$ for $x, y \sim \mathcal{D}$, $t \sim \widehat{\mathcal{T}}$, and $t' \sim \widehat{\mathcal{T}}|t$. Here, $\mathcal{D}$ represents the input-label data distribution, $\widehat{\mathcal{T}}$ denotes the augmentation distribution for extracting the *symbol* from $x$, and $\widehat{\mathcal{T}}|t$ represents the augmentation distribution for extracting the context of $\sigma$. Our goal is to minimize this variation of the InfoNCE loss between the encoder $E$ and the embedding function $\mathcal{E}$:

$$\mathcal{L}_{NCE}(D) = -\frac{1}{N} \sum_{i=1}^{N} \log \left( \frac{\exp(\text{sim}(\mathcal{E}(\sigma_i), E(\rho_i, y_i)))}{\sum_j \exp(\text{sim}(\mathcal{E}(\sigma_i), E(\rho_j, y_j)))} \right) \tag{2}$$

The effect of this loss is to encourage the embedding function $\mathcal{E}$ to encode symbol $\sigma_i$ close to their corresponding context-label pairs $\rho_i, y_i$ and far apart from the noisy pairs $\rho_j, y_j$, for $j \neq i$. This fact can be seen by simply deriving the update rule for the embedding function $\mathcal{E}$ and the encoder function $E$. This is provided in Appendix A.5.

Further, it can be shown that, an optimal model, made of $\mathcal{E}^*$ and $E^*$, for $\mathcal{L}_{NCE}$ needs to encode semantically equivalent symbols with exactly the same vector. This is the subject of the following theorem:

**Theorem 3.1.** *Given the symbols $u, v \in \Sigma$ such that 1. $u \stackrel{\circ}{=} v$, 2. $\forall \rho : p(\rho|u) = p(\rho|v)$, and 3. if there are $(\rho_1, y_1), \ldots, (\rho_d, y_d)$ context-label pairs such that $E^*(\rho_i, y_i)$ form a basis for $\mathbb{R}^d$ then $\mathcal{E}^*(u) = \mathcal{E}^*(v)$*

On one hand, semantic equivalence relations do not depend on the conditional distribution ($p(\rho|u) = p(\rho|v)$); therefore, these hypotheses may seem unnecessary. On the other hand, the data must still reflect these semantic equivalence relations to some extent. For instance, consider an edge case where two symbols are semantically equivalent, but one of them never appears in the dataset. In this scenario, it is natural that we would be unable to discover this semantic equivalence relation.

The final hypothesis, concerning the basis, ensures that the only solution is $\mathcal{E}^*(u) = \mathcal{E}^*(v)$. Given that most realistic scenarios encompass thousands of possible context-label pairs while the embedding size typically does not exceed $d = 1024$. Further, it should be noted that establishing a basis for $\mathbb{R}^d$ only requires $d$ linearly independent embeddings. The determinant forms a polynomial of the entries $E^*(\rho_1, y_1), \ldots, E^*(\rho_d, y_d)$ and has a Lebesgue measure of zero. Thus, slight perturbation of the encoded vectors would establish a basis almost surely. This hypothesis is employed in the context of invertible neural networks Finzi et al. (2019), where assuming a weight matrix to be a basis implies non-singularity, allowing for inversion. Additionally, one can always reduce the embedding size to ensure that this hypothesis holds. Ultimately, this hypothesis can be considered realistic for most use cases.

Please refer to Appendix A.2 for a formal proof of Theorem 3.1.

## 4 CONTRASTIVE LEARNERS ARE SEMANTIC LEARNERS

As we have seen, the notion of semantic equivalence is strongly tied to a label distribution. However, the rise of contrastive learning is partially due to the fact that it does not require labels (which are usually expensive to obtain) to generate embeddings organized in a semantic structure (Gao et al. (2021); Le-Khac et al. (2020); He et al. (2020)). This raises a natural question: how do contrastive learning techniques learn formal semantics without access to labels?

To answer this question, we need to formalize the concepts of downstream and contrastive tasks. Let us denote with $\mathcal{D}$ the data distribution for the downstream task. We also assume to be able to sample symbol-context-label triplets, $\sigma, \rho, y \sim \mathcal{D}$. Further, we will denote with $p_{\mathcal{D}}(y|\sigma, \rho)$ the probability of label $y$ given the symbol-context pair $\sigma$ and $\rho$.

Now that the downstream task has been defined, we proceed to the contrastive task. Here, we are given a data distribution $\mathcal{C}$. From this distribution, we can only sample context-symbol pairs, $\sigma, \rho \sim \mathcal{C}$ but we have no access to a label.

Of course, we need to realize that we cannot apply contrastive learning to any data distribution and hope to learn useful representations for a downstream task of interest. For example, we cannot hope to learn useful representations for an animal classifier from a white noise distribution. Therefore, we will need some kind of hypothesis between contrastive and downstream data to hold. We refer to this hypothesis as the distributional alignment hypothesis:

**Definition 4.1** (Distributional Alignment Hypothesis)**.** A downstream distribution $\mathcal{D}$ and a contrastive distribution $\mathcal{C}$ are aligned iff.

$$\forall y \in \mathcal{Y}, \rho \in \text{P} : p_{\mathcal{D}}(y|u, \rho) = p_{\mathcal{D}}(y|v, \rho) \iff \forall \rho \in \text{P} : p_{\mathcal{C}}(\rho|u) = p_{\mathcal{C}}(\rho|v)$$

On the left, we state that $u$ and $v$ are semantically equivalent according to the downstream task. On the right, we state that $u$ and $v$ are conditionally equivalent for the contrastive task. Notice that, this definition highly resembles the definition of distributional hypothesis as formalized by Bertolotti & Cazzola (2024). The main and only difference between the distributional hypothesis and the distributional alignment hypothesis is simply that the first is concerned with a single data distribution while the latter is concerned with two distributions (one for the downstream data and one for the contrastive data).

Now, suppose we aim to minimize the loss in Equation 2 without access to the labels. Then we minimize the following:

$$\mathcal{L}_{NCE}(D) = -\frac{1}{N} \sum_{i=1}^{N} \log \left( \frac{\exp(\text{sim}(\mathcal{E}(\sigma_i), E(\rho_i)))}{\sum_j \exp(\text{sim}(\mathcal{E}(\sigma_i), E(\rho_j)))} \right)$$

One can now show that the following theorem holds:

**Theorem 4.2.** *Given the symbols $u, v \in \Sigma$ such that: 1. $\forall \rho : p(\rho|u) = p(\rho|v)$. 2. If there are $\rho_1, \ldots, \rho_d$ contexts such that $E^*(\rho_i)$ form a basis for $\mathbb{R}^d$ then $\mathcal{E}^*(u) = \mathcal{E}^*(v)$*

This theorem shows that an optimal embedding function needs to encode such symbols $u$ and $v$ with the same vector. The proof of this theorem is provided in Appendix A.3.

Similar argument discussed for Theorem 3.1 can be applied here. The hypothesis concerning the basis is necessary to ensure that the only solution is $\mathcal{E}^*(u) = \mathcal{E}^*(v)$, and it can be considered realistic. The $p(\rho|u) = p(\rho|v)$ hypothesis, despite being more strong, is necessary to ensure that the data reflects the semantic equivalence relations.

By combining Definition 4.1 and Theorem 4.2, it is easy to derive the following corollary:

**Corollary 4.3.** *1. Let $\mathcal{C}$ and $\mathcal{D}$ be a distributionally aligned contrastive and downstream distributions, respectively. 2. Let $\mathcal{E}^*$ and $E^*$ be the optimal embedding and encoder functions for the contrastive task. 3. Let $\rho_1, \ldots, \rho_d$ contexts such that $E^*(\rho_i)$ form a basis for $\mathbb{R}^d$. Then,*

$$u \stackrel{\circ}{=}_\mathcal{D} v \iff \mathcal{E}^*(u) = \mathcal{E}^*(v)$$

Here, $u \stackrel{\circ}{=}_\mathcal{D} v$ denotes the semantic equivalence relation according to the downstream task $\mathcal{D}$. The proof of this corollary is immediate. Firstly, from the distributional alignment hypothesis (Definition 4.1), we have that, two symbols are semantically equivalent for the downstream task if and only if they are conditionally equivalent for the contrastive task. Further, two conditionally equivalent symbols are encoded with the same vector by the optimal embedding and encoder functions for the contrastive task (Theorem 4.2). Therefore, two symbols are encoded within the same vector only if they were semantically equivalent to begin with. The backward implication is derived in Proposition A.2 of the Appendix Section A.4.

This result highlight a fundamental connection between contrastive learning and semantic learning (we refer to semantic learning as the process to learn formal semantic relations). In particular, it shows that contrastive learning can be seen as a form of semantic learning, where the semantic equivalence relations (precisely those state in Definition 2.1) are learned from the data distribution itself.

| $a$ | $b$ | $y$ |
|-----|-----|-----|
| 45 | 5 | 0 |
| 12 | 21 | 3 |
| 66 | 12 | 8 |

Table 1: Examples of downstream data for the ModAdd task. $a, b, c \in \{0, \ldots, 100\}$ and $k = 10$.

| $a$ | $b$ | $c$ | $y$ |
|-----|-----|-----|-----|
| 45 | ? | 0 | 5 |
| 12 | ? | 3 | 21 |
| ? | 12 | 8 | 66 |

Table 2: Examples of contrastive data from the CModAdd task. $a, b \in \{0, \ldots, 100, ?\}$ and $k = 10$.

## 5 EXPERIMENTS

In this section, we will focus on designing a small controlled experiment where the hypothesis of the previous theorem can be easily verified.

**Modular Addition Task.** We will consider an algorithmic task commonly referred to as modular addition, or ModAdd for short (Power et al. (2022); Gromov (2023); Furuta et al. (2024)). In ModAdd, a model is tasked with solving simple modular addition equations of the form:

$$(a + b) \mod k = y$$

where $y$ represents the unknown quantity, and $a, b, k \in \{0, \dots, N\}$ with $N \in \mathbb{N}$. Typically, $k$ is fixed at the start of training, and the neural network (NN) is presented with pairs of natural numbers and tasked with predicting the correct modulo class. A few training examples are provided in Table 1.

**Semantic Equivalence in ModAdd.** It should be apparent that all symbols of the form $a + ik$ are semantically equivalent to $a$ in ModAdd. For example, consider the problem instantiated with $k = 10$ and $N = 100$. In this case, symbols like $0, 10, 20, \dots, 100$ are semantically equivalent. In other words, if we swap 10 with 80, we do not alter the outcome distribution of the problem (i.e., $\forall y, a : p(y \mid a + 80 \mod 10) = p(y \mid a + 10 \mod 10)$).

**Contrastive Modular Addition Task.** The contrastive modular addition task, or CModAdd for short, is a simple extension of the previous ModAdd task. In CModAdd, we generate sequences of three symbols. The first symbol represents the first addend, $a$, which is sampled according to $a \sim \mathcal{U}(0, N)$. The second symbol represents the second addend, $b$, sampled according to $b \sim \mathcal{U}(0, N)$. The third symbol represents the result of the modulus operation, $c = (a + b) \mod k$. Additionally, we randomly mask one of the symbols $a$ or $b$, and refer to the masked symbol as $y$. The goal of this task is to train a representation for the symbols that can be useful for the downstream task ModAdd. Table 2 provides a few examples for clarity.

**Conditional Equivalence in CModAdd.** It should be apparent that all symbols of the form $a + ik$ are conditionally equivalent to $a$ in CModAdd. For example, consider the problem instantiated with $k = 10$ and $N = 100$. In this case, symbols like $0, 10, 20, \dots, 100$ are conditionally equivalent. In other words, the probability of seeing 10 under the mask token is always the same as the probability of seeing 80 under the mask token (i.e., $\forall y, a : p(a+? \mod 10 = y \mid 10) = p(a+? \mod 10 = y \mid 80)$).

**CModAdd and ModAdd are aligned.** Let $u$ and $v$ be semantically equivalent symbols according to ModAdd (i.e., $\forall y, a : p(y \mid a + u \mod k) = p(y \mid a + v \mod k)$). Then, it is easy to see that $u$ and $v$ are conditionally equivalent for CModAdd ($\forall y, a : p(a+? \mod k = y \mid u) = p(a+? \mod k = y \mid v)$) and vice versa. This property aligns exactly with Definition 4.1. Therefore, we can say that CModAdd and ModAdd are distributionally aligned. Furthermore, by applying Theorem 4.2, we find that employing a simple contrastive procedure will result in conditionally equivalent symbols for CModAdd being encoded into spatially close vectors. Since conditionally equivalent symbols in CModAdd are semantically equivalent in ModAdd, we can conclude that semantically equivalent symbols in ModAdd will be encoded into spatially close embeddings as stated from Corollary 4.3.

**ModAdd & CModAdd Datasets.** We generated all possible samples for a ModAdd and CModAdd problem with $N = 16$ and $k = 8$. Of this dataset, only the $80\%$ is used for training purposes, the other $20\%$ is used as a validation set to perform periodic evaluation (one each 10 epochs).

**CModAdd Architecture.** The CModAdd architecture consists of two components: 1. an embedding function, and 2. an encoder function. As previously mentioned, the embedding function maps symbols to their $d$-dimensional representations. The encoder function, on the other hand, maps contexts to their $d$-dimensional representations. The embedding function is simply a parameter matrix of size $N \times d$, mapping each symbol index to a randomly initialized $d$-dimensional vector. The encoder function consists of: 1. an embedding layer, 2. a 3-layer transformer encoder, and 3. a linear layer that maps the latent space to the $d$-dimensional representation. To comply with the third hypothesis of Theorem 4.2, $d$ is chosen to be small, specifically $d = 8$.

**ModAdd Architecture.** The ModAdd architecture consists of: 1. an embedding layer, 2. a 3-layer transformer encoder, and 3. a linear layer that maps the latent space to the label. Again, the embed-

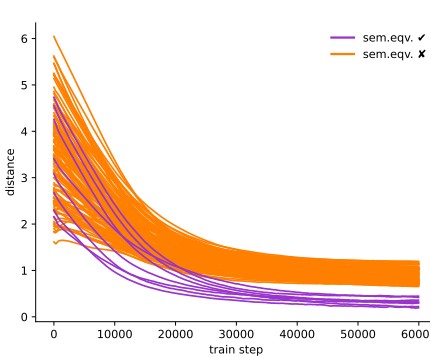

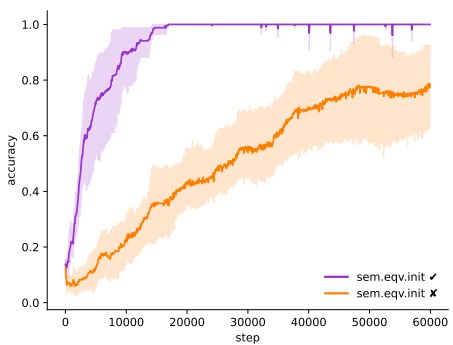

(a) Symbol Embedding distances between semantically equivalent (purple) pairs and semantically different pairs (orange) measured at different training steps

(b) Validation accuracy (mean and 95% confidence interval) of 5 models measured at different training steps. In purple, a model that is initialized with SimCLR embeddings. In orange, the same model but with random embedding initialization

Figure 2: Embedding distances for the CModAdd task and validation accuracy for the ModAdd task with and without pre-trained embeddings

ding size is chosen to be small, $d = 8$. This allows for initializing these embeddings with those trained from the CModAdd architecture.

**Training Hyperparameters.** The training proceed for $1e4$ epochs with Adam optimizer (Kingma & Ba (2015)). The learning rate is set to $1e-4$ and we employ weight decay with value $1e-2$. The batch size is set to 32.

**Contrastive Results.** The results are shown in Figure 2a. In this figure, we plot how the Euclidean distance between different symbol embeddings evolves during training. The orange line represents symbol pairs that are not semantically equivalent, while the purple line represents symbol pairs that are semantically equivalent. As observed, during training, only semantically equivalent symbols become close to each other, while the distance between semantically different pairs does not converge beyond a certain level. We note that the distance does not become zero (as the theory would suggest) this is likely due to the loss becoming close to zero preventing noticeable movement between the embeddings.

**Classification Results.** The results are shown in Figure 2b. To simulate a common procedure involving contrastive pre-training followed by classification training, we first train an embedding function by minimizing the loss shown in Equation 2 On the CModAdd data. Next, we train a classification model for the ModAdd task. The orange line represents the results for a randomly initialized model, while the purple line represents the results for a model initialized with the pre-trained embeddings. The model initialized with pre-trained embeddings achieves better results earlier in training compared to the randomly initialized model.

In Section A.6, we present additional experiments that vary, model size and training procedure.

## 6 DISCUSSION

Let us recall that the main goal of contrastive learning is to learn *good* representations of data for downstream tasks, which often have limited availability. But what constitutes a good representation? One might be tempted to define it as a representation that facilitates fast and efficient learning of the downstream task. However, this definition doesn't specify any particular properties that the representation should possess. As a result, questions like **When is a contrastive dataset good for a specific downstream task?** remain unanswered.

We argue that good representations should, at a minimum, encode semantic equivalence relations (Definition 2.1). In other words, if two symbols $u$ and $v$ are semantically equivalent (as defined in

Definition 2.1), their representations should be close to each other, meaning they should be encoded as vectors in close proximity. Conversely, non-equivalent symbols should have representations that are further apart. This belief is strongly supported by Figure 2b, where a model initialized with semantic equivalence relations achieves perfect accuracy early in training compared to the randomly initialized one.

If we accept that good representations need to encode semantic equivalence relations, the next question is: **how do we train embeddings that encode these equivalence relations?** To address this, we turn to a popular pre-training method—contrastive learning. Empirically, contrastive learning has often been observed to generate semantically meaningful relations (Wu et al. (2018); Ge et al. (2022)), although semantics is rarely discussed formally. If we accept the definition of semantic as the one presented in Definitions 2.1, we notice that it heavily depends on the presence of labels. Now, if we imagine feeding labels into a contrastive procedure, one could show that a model minimizing Equation 2 would need to encode semantic equivalence relations. This is the subject of Theorem 3.1. However, contrastive learners usually do not have access to labels. In such cases, it can be shown that when the contrastive data are aligned with the downstream data (Definition 4.1), the trained representation must encode these equivalence relations for the downstream task. This is the subject of Corollary 4.3. This behavior is also empirically demonstrated in Figure 2a, where, during contrastive training, embeddings of semantically equivalent symbols converge, while embeddings of semantically different symbols do not.

To summarize the results of this work let us answer briefly the proposed research question:

**Under what conditions is a contrastive task useful for a downstream task?**

*A contrastive dataset can be used for a downstream task when the distributional alignment hypothesis holds.*

**How can we train embeddings that effectively encode semantical equivalence relations?**

*The optimal SimCLR variant (under the proper hypotheses), discussed in Section 3, is formally guaranteed to encode semantic equivalence relations. Empirically, the research community has observed this behavior in a variety of different settings.*

## 7    THREATS TO VALIDITY

In this section, we aim to address the limitations and potential threats that could impact our conclusions.

**External Validity**    We claimed that the distributional alignment hypothesis is crucial to the success of contrastive learning. We supported this claim with both empirical and theoretical evidence. However, it should be noted that, in most realistic scenarios, verifying this hypothesis is impractical. Consequently, in real-world applications, the achieved performance on downstream tasks remains the only empirical measure of the quality of the representation.

**Internal Validity**    While Theorems 3.1, and 4.2 formalize the intuitions of this work, they also rely on somewhat strong hypotheses that limit their applicability to most real-world scenarios.

One common assumption in this manuscript is the conditional equivalence $\forall \rho : p(\rho|u) = p(\rho|v)$. This assumption is often too strong for practical applications. However, it can be justified by the fact that if the symbols $u$ and $v$ never appear in the same context, discovering their semantic equivalence would be impossible. Nevertheless, it is important to note that contrastive learners are often capable of encoding semantic similarities even when this assumption is violated to some extent. This suggests that the results of this work could be generalized by relaxing the conditional equivalence assumption.

The alignment hypothesis (Definition 4.1) used in the proof of Corollary 4.3 is quite strong. In practice, the alignment between representations of semantically equivalent symbols is often only partial, yet contrastive learning has shown success even in such cases. Nevertheless, the conditions under which a contrastive dataset can effectively support a downstream task remain very stringent. For example, randomly generated data is unlikely to be useful for any downstream task, and genomic

contrastive data would provide little value to a natural language sentiment classifier. Ultimately, this suggests that strong hypotheses are necessary even in practical applications.

## 8   RELATED WORKS

**Contrastive Learning in practice.**     Contrastive learning is one of the most popular and studied pre-training techniques. While we present a fairly general framework, other works focus on specific data modalities. For example, contrastive learning techniques for text sentences are discussed by Gao et al. (2021) and Aberdam et al. (2021), see Xu et al. (2023) for a review. On source code contrastive learning has been applied by Bertolotti & Cazzola (2023); Wang et al. (2022b) and Ding et al. (2023). You et al. (2020); Zhu et al. (2020), and Qiu et al. (2020) focus on graph representations, see Liu et al. (2022) for a review. In particular, the work of Xia et al. (2022) focuses on a framework that does not require augmentation. Meanwhile, the work of Wang et al. (2020); Park et al. (2020) and Wang et al. (2023) focuses on images. Tsai et al. (2021a) combine a mixture of expert techniques with contrastive learning image representations. Several long-term forecasting problems are addressed by Park et al. (2024). Other works focus on multiple data modalities (Daunhawer et al. (2023) and Yuan et al. (2021)), see Zong et al. (2024) for a recent survey. Lo et al. (2024) applied contrastive learning to the agent communication problem. Refer to Rani et al. (2023) and Gui et al. (2024) for general survey regarding self-supervised learning and contrastive learning.

**Contrastive Learning Techniques.**     While we mainly focus on the InfoNCE loss with SimCLR. However, there are several other contrastive objectives and technique that have been proven useful in practice that we do not address in the previous sections. Cho (2005) propose one of the first contrastive objectives introduced in the literature that aims to maximize similarity from pairs of the same class and minimize otherwise. A similar objective in principle is the triple loss (Schroff et al. (2015)). An extension to the triple loss, known as Lifted Structured Loss is proposed by Oh Song et al. (2016). NCE and InfoNCE losses developed by Gutmann & Hyvärinen (2010) and van den Oord et al. (2018) exploit the concept of matching a data point with the correct one among a set of noise data points. More recently, Shidani et al. (2024) proposes a multi-view objective. Lu et al. (2022) tackles the out-of-distribution problem proposing invariant Causal Representation Learning—iCaRL. Zero-CL (Zhang et al. (2022)) is a contrastive learning technique that does not rely on negative pairs. Wang et al. (2022a) develops PiCO, a contrastive learning framework for partial label learning. Zhang et al. (2023) proposes a contrastive loss based distributionally robust optimization. The work of Fort et al. (2021); Ge et al. (2021); Wen & Li (2021) focuses on studying and designing the effect multiple and different augmentations.

**Understanding Contrastive Learning.**     In their works, Arora et al. (2019); Lee et al. (2021) present some of the first theoretical analyses of contrastive learning, establishing a connection between the contrastive task and the downstream task. Similarly, HaoChen et al. (2021) and Wang et al. (2022c) relax some of the strong assumptions made in earlier analyses. Their work is based on the existence of a latent variable from which positive and negative contrastive data pairs can be sampled. The work of Saunshi et al. (2022) challenge prior theoretical result in scenarios with different inductive biases. Wang & Isola (2020) prove that cross-entropy contrastive loss optimizes closeness and uniformity. An information-theoretic perspective is provided by Tian et al. (2020); Tsai et al. (2021b); Tosh et al. (2021), focusing on learning sufficient statistics for the downstream task. Johnson et al. (2022) unveils a connection between contrastive learning and kernel PCA (Schölkopf et al. (1997)). Alon et al. (2023) focus on providing a bound on the sample complexity for several contrasting learning scenarios. Simon et al. (2023) study the stepwise learning phenomenon observed in contrastive learners through the lenses of linearized model. The work of Shen et al. (2022) and Garg et al. (2024) compares these techniques when distributional shifts occur. Cui et al. (2023) focuses on contrastive learning when joined with weak supervision. The works of Tschannen et al. (2019); Poole et al. (2019); Lee et al. (2024); Gálvez et al. (2023) study the relation between contrastive learning and mutual information. In this line of work, Levy & Goldberg (2014) shows that skip-grams with negative sampling is equivalent to perform matrix factorization. Wu et al. (2023) offers a perspective from distributionally robust optimization theory. Compared to these works, our analysis stems from the notion of semantic equivalence and explores under what conditions this relationship can be learned using contrastive data.

## 9 CONCLUSION

In this work, we analyzed what constitutes a good contrastive task from the theoretical perspective of semantics. In Sections 3 and 4 we formalized the conditions under which contrastive learning can effectively encode semantics equivalence relations in both scenarios—with and without labels available. In the latter scenario, we found that the distributional alignment hypothesis (Definition 4.1) plays a fundamental role. Furthermore, we empirically verified the intuitions formalized earlier in Section 5 through small, controlled experiments where all hypotheses could be easily tested. To further validate the presented intuitions, we conducted additional experiments discussed in Appendix A.6. The code to reproduce the experiments is freely available at the following URL:

$$\texttt{https://redacted-for-anonimity.com/}$$

## 10 FUTURE WORK

In this work, we have shown that an optimal contrastive model must learn to encode semantic equivalence relations under the distributional alignment hypothesis (Definition 4.1) and ideal assumptions. However, in practical scenarios, the alignment between contrastive data and downstream tasks is often partial, and the conditional equivalence assumption is frequently violated. A potential direction for future research is to explore the robustness of the proposed theory when these assumptions are only partially met or entirely invalid.

Multitask learning is a paradigm focused on training models to perform multiple tasks simultaneously. In the work of Maurer et al. (2016), the learning model is structured with a shared representation function across tasks and a task-specific classification (or regression) head for each task. We hypothesize that the shared representation function should encode semantic equivalence relations across all tasks concurrently. However, an intriguing question arises: what happens when the semantic structure is not well-aligned across tasks? Addressing this issue presents a compelling avenue for future work.

ACKNOWLEDGMENTS

Redacted for Anonimity

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

## A APPENDIX

In these appendices, we present the proofs for Theorems 3.1 and 4.2 in Sections A.2 and A.3, respectively. Additionally, we provide details on the updates to the embedding function $\mathcal{E}$ in Section A.5, followed by a series of supplementary experiments in Section A.6.

## A.1 PRELIMINARIES

Firstly, let us begin with a Lemma that will be proven useful in later sections.

**Lemma A.1.** *Let $\mathcal{E}^*, E^*$ be the optimal embedding and encoder functions such that for an unconstrained hypothesis $\mathcal{H} = \{(\mathcal{E}, E) | \mathcal{E} : \Sigma \to \mathbb{R}^d, E : \mathrm{P}, \mathcal{Y} \to \mathbb{R}^d\}$ we minimize the conditional risk associated with the infoNCE loss:*

$$\mathcal{E}^*, E^* = \arg\min_H \left\{ \mathbb{E}_{i|y,\rho,\sigma_1,\ldots,\sigma_n} \left[ -\log \left( \frac{\exp(\mathrm{sim}(\mathcal{E}(\sigma_i), E(y, \rho)))}{\sum_j \exp(\mathrm{sim}(\mathcal{E}(\sigma_i), E(y, \rho)))} \right) \right] \right\}$$

*Where $i$ represents the random variable that associates the context-label pair $(\rho, y)$ with the correct symbol from $\sigma_1, \ldots, \sigma_n$.*

*Then*

$$\forall \rho, y, \sigma_1, \ldots, \sigma_n : p(\sigma_i | \rho, y) = \exp(\mathrm{sim}(\mathcal{E}^*(\sigma_i), E^*(y, \rho))) p(\sigma_i) \sum_j \frac{p(\sigma_j | \rho, y)}{p(\sigma_j)}$$

*Proof.* Let us begin by considering the conditional risk associated with the infoNCE loss:

$$\mathcal{R}(\mathcal{E}, E, \mathcal{D}) = \mathbb{E}_{i|y,\rho,\sigma_1,\ldots,\sigma_n} \left[ -\log \left( \frac{\exp(\mathrm{sim}(\mathcal{E}(\sigma_i), E(y, \rho)))}{\sum_j \exp(\mathrm{sim}(\mathcal{E}(\sigma_j), E(y, \rho)))} \right) \right]$$

For brevity, let us denote $f_i$ as the term $\exp(\mathrm{sim}(\mathcal{E}(\sigma_i), E(\rho, y)))$. Then, we obtain:

$$\mathcal{R}(\mathcal{E}, E, \mathcal{D}) = \mathbb{E}_{i|y,\rho,\sigma_1,\ldots,\sigma_n} \left[ -\log \left( \frac{f_i}{\sum_j f_j} \right) \right]$$

It should be fairly easy to see that this term amounts to a cross-entropy term between a one-hot distribution, $e_i$, and the predicted distribution $f = \left[ \frac{f_0}{\sum_j f_j}, \ldots, \frac{f_n}{\sum_j f_j} \right]$:

$$\mathcal{R}(\mathcal{E}, E, \mathcal{D}) = \mathbb{E}_{i|y,\rho,\sigma_1,\ldots,\sigma_n} \left[ -e_i^T \log(f) \right] = \mathbb{E}_{i|y,\rho,\sigma_1,\ldots,\sigma_n} [H(e_i, f)]$$

It is a standard result that the probability distribution minimizing the conditional expectation of the cross-entropy, $\arg\min_f \mathbb{E}_{y|x}[H(e_y, f)]$, is $p(y|x)$ Hastie et al. (2009). Applying this result to our context, we conclude that:

$$f_i^* = p(i | \rho, y, \sigma_1 \ldots, \sigma_n)$$

Furthermore, as shown in van den Oord et al. (2018):

$$\begin{aligned}
p(i | \rho, y, \sigma_1, \ldots, \sigma_n) &= \frac{p(\sigma_i | \rho, y) \prod_{k \neq i} p(\sigma_k)}{\sum_j p(\sigma_j | \rho, y) \prod_{k \neq j} p(\sigma_k)} \\
&= \frac{\frac{p(\sigma_i | \rho, y)}{p(\sigma_i)} p(\sigma_i) \prod_{k \neq i} p(\sigma_k)}{\sum_j \frac{p(\sigma_j | \rho, y)}{p(\sigma_j)} p(\sigma_j) \prod_{k \neq j} p(\sigma_k)} \\
&= \frac{\frac{p(\sigma_i | \rho, y)}{p(\sigma_i)} \prod_k p(\sigma_k)}{\sum_j \frac{p(\sigma_j | \rho, y)}{p(\sigma_j)} \prod_k p(\sigma_k)} \\
&= \frac{\frac{p(\sigma_i | \rho, y)}{p(\sigma_i)}}{\sum_j \frac{p(\sigma_j | \rho, y)}{p(\sigma_j)}}
\end{aligned}$$

Therefore, we have:

$$\frac{\frac{p(\sigma_i|\rho,y)}{p(\sigma_i)}}{\sum_j \frac{p(\sigma_j|\rho,y)}{p(\sigma_j)}} = p(i|\rho,y,\sigma_1,\ldots,\sigma_n) \implies$$

$$\frac{\frac{p(\sigma_i|\rho,y)}{p(\sigma_i)}}{\sum_j \frac{p(\sigma_j|\rho,y)}{p(\sigma_j)}} = f_i^* \implies$$

$$p(\sigma_i|\rho,y) = f_i^* p(\sigma_i) \sum_j \frac{p(\sigma_j|\rho,y)}{p(\sigma_j)} \implies$$

$$p(\sigma_i|\rho,y) = \exp(\mathrm{sim}(\mathcal{E}^*(\sigma_i), E^*(y,\rho)))p(\sigma_i) \sum_j \frac{p(\sigma_j|\rho,y)}{p(\sigma_j)}$$

Notice that we made no assumptions about $\rho$, $y$, or $\sigma_1,\ldots,\sigma_n$, except that one symbol is distributed conditionally on the context-label pair, while the others are not. Therefore, we conclude that the statement of the lemma holds for any choice of these variables.

$\square$

### A.2 THEOREM 3.1

Let us consider the first theorem:

**Theorem.** *Given the symbols $u,v \in \Sigma$ such that*

1. *$u \overset{\circ}{=} v$,*

2. *$\forall \rho : p(\rho|u) = p(\rho|v)$, and*

3. *if there are $(\rho_1,y_1),\ldots,(\rho_d,y_d)$ context-label pairs such that $E^*(\rho_i,y_i)$ form a basis for $\mathbb{R}^d$*

*then $\mathcal{E}^*(u) = \mathcal{E}^*(v)$*

This theorem states the condition for SimCLR to encode semantically equivalent symbols into equivalent embeddings.

*Proof.* Since $u \overset{\circ}{=} v$, (i.e. $\forall \rho : p(y|u,\rho) = p(y|v,\rho)$), we can write:

$$p(u|y,\rho) = \frac{p(u,y,\rho)}{p(y,\rho)} = \frac{p(y|u,\rho)p(u,\rho)}{p(y,\rho)} = \frac{p(y|v,\rho)p(u,\rho)}{p(y,\rho)} = \frac{p(y,v,\rho)p(u,\rho)}{p(y,\rho)p(v,\rho)} = \frac{p(v|y,\rho)p(u,\rho)}{p(v,\rho)}$$

Or,

$$p(u|y,\rho)p(v,\rho) = p(v|y,\rho)p(u,\rho)$$
$$p(u|y,\rho)p(\rho|v)p(v) = p(v|y,\rho)p(\rho|u)p(u)$$
$$(*)\ \exp(\mathrm{sim}(\mathcal{E}^*(u), E^*(y,\rho)))p(u)p(\rho|v)p(v) = \exp(\mathrm{sim}(\mathcal{E}^*(v), E^*(y,\rho)))p(v)p(\rho|u)p(u)$$
$$\exp(\mathrm{sim}(\mathcal{E}^*(u), E^*(y,\rho)))p(\rho|v) = \exp(\mathrm{sim}(\mathcal{E}^*(v), E^*(y,\rho)))p(\rho|u)$$

The step $(*)$ can be concluded by considering Lemma A.1. Take a context-label pair $\rho, y$ sampled with symbols $\sigma_1,\ldots,\sigma_N$, one of which will be $u$ and one of which will be $v$. Further, let $i_u$ and $i_v$ be the indices of $u$ and $v$ in the set of symbols. By Lemma A.1, we have:

$$p(u|\rho,y) = \exp(\mathrm{sim}(\mathcal{E}^*(u), E^*(y,\rho)))p(u)\left(\frac{p(u|\rho,y)}{p(u)} + \frac{p(v|\rho,y)}{p(v)} + \sum_{\substack{j \\ j \neq i_u, i_v}} \frac{p(\sigma_j|\rho,y)}{p(\sigma_j)}\right)$$

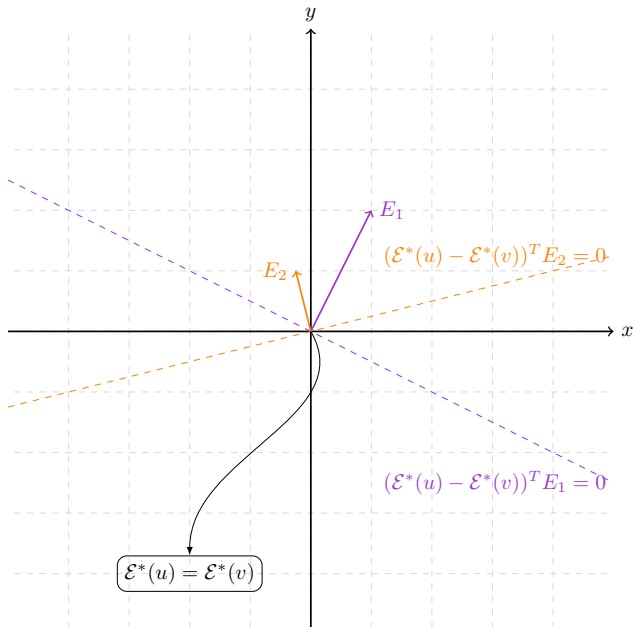

Figure 3:
2−dimensional representation of: $(\mathcal{E}^*(u) - \mathcal{E}^*(v))^T E_1 = 0$ (purple) and $(\mathcal{E}^*(u) - \mathcal{E}^*(v))^T E_2 = 0$ (orange). $E_1$ and $E_2$ have to form a basis of $\mathbb{R}^2$ in order for the solution to be unique.

A similar equation can be written for $v$:

$$p(v|\rho, y) = \exp(\text{sim}(\mathcal{E}^*(v), E^*(y, \rho)))p(v) \left( \frac{p(u|\rho, y)}{p(u)} + \frac{p(v|\rho, y)}{p(v)} + \sum_{\substack{j \\ j \neq i_u, i_v}} \frac{p(\sigma_j|\rho, y)}{p(\sigma_j)} \right)$$

Since both summation term are equal, in equation $(*)$ they can be canceled out. Further, me made no assumption from the context-label pair $\rho, y$. Therefore, by applying the hypothesis $p(\rho|u) = p(\rho|v)$ we can write:

$$\forall \rho, y : \exp(\mathcal{E}^*(u)^T E^*(y, \rho)) = \exp(\mathcal{E}^*(v)^T E^*(y, \rho))$$

Next consider $(\rho_1, y_1), \ldots, (\rho_d, y_d)$ context-label pairs such that $E^*(y_i, \rho_i) = E_i$ form a basis of $\mathbb{R}^d$:

$$\begin{cases} \exp(\mathcal{E}^*(u)^T E_1) = \exp(\mathcal{E}^*(v)^T E_1) \implies (\mathcal{E}^*(u) - \mathcal{E}^*(v))^T E_1 = 0 \\ \ldots \\ \exp(\mathcal{E}^*(u)^T E_d) = \exp(\mathcal{E}^*(v)^T E_d) \implies (\mathcal{E}^*(u) - \mathcal{E}^*(v))^T E_d = 0 \end{cases}$$

The above system has $d$ equations stating that $\mathcal{E}^*(u) - \mathcal{E}^*(v)$ is perpendicular to every basis vector $E_0, \ldots, E_d$. Thus, Only one solution is possible: $\mathcal{E}^*(u) = \mathcal{E}^*(v)$. Fig. 3 provides a graphical representation of the system when $d = 2$. In this case, we have only two equations whose solutions lie onto a line. The solution is unique when $E_1$ and $E_2$ are linearly independent (i.e., form a basis for $\mathbb{R}^2$). □

### A.3 THEOREM 4.2

Let us consider the first theorem:

**Theorem.** *Given the symbols $u, v \in \Sigma$ such that:*

1. $\forall \rho : p(\rho|u) = p(\rho|v)$.

2. *If there are $\rho_1, \ldots, \rho_d$ contexts such that $E^*(\rho_i)$ form a basis for $\mathbb{R}^d$*

*then $\mathcal{E}^*(u) = \mathcal{E}^*(v)$*

This theorem states the condition for our approach to encode conditionally equivalent symbols into equivalent embeddings. For completeness, we provide the complete proof to this theorem. However, this is an almost exact match to the proof of Theorem A.2.

*Proof.* Let us begin by considering the hypothesis: $p(\rho|u) = p(\rho|v)$:

$$\forall \rho : p(\rho|u) = p(\rho|v) \implies$$
$$\forall \rho : \frac{p(\rho, u)}{p(u)} = \frac{p(\rho, v)}{p(v)} \implies$$
$$\forall \rho : \frac{p(u|\rho)p(\rho)}{p(u)} = \frac{p(v|\rho)p(\rho)}{p(v)} \implies$$
$$(*)\forall \rho : \exp(\mathcal{E}^*(u)^T E^*(\rho)) = \exp(\mathcal{E}^*(v)^T E^*(\rho))$$

The step $(*)$ follows, once again, from an application of Lemma A.1. Next consider $\rho_1, \ldots, \rho_d$ contexts such that $E^*(\rho_i) = E_i$ form a basis of $\mathbb{R}^d$:

$$\begin{cases} \exp(\mathcal{E}^*(u)^T E_1) = \exp(\mathcal{E}^*(v)^T E_1) \implies (\mathcal{E}^*(u) - \mathcal{E}^*(v))^T E_1 = 0 \\ \ldots \\ \exp(\mathcal{E}^*(u)^T E_d) = \exp(\mathcal{E}^*(v)^T E_d) \implies (\mathcal{E}^*(u) - \mathcal{E}^*(v))^T E_d = 0 \end{cases}$$

The above system has $d$ equations stating that $\mathcal{E}^*(u) - \mathcal{E}^*(v)$ is perpendicular to every basis vector $E_0, \ldots, E_d$. Thus, Only one solution is possible: $\mathcal{E}^*(u) = \mathcal{E}^*(v)$. Fig. 3 provides a graphical representation of the system when $d = 2$. In this case, we have only two equations whose solutions lie onto a line. The solution is unique when $E_1$ and $E_2$ are linearly independent (i.e., form a basis for $\mathbb{R}^2$). $\qquad\square$

A.4 PROPOSITION A.2

Now, let us introduce a discuss the proposition for the backward implication of the Corollary 4.3.

**Proposition A.2.** *Given symbols $u$ and $v$ such that $\forall \rho : p(\rho|u) = p(\rho|v)$. Given optimal embedding function $\mathcal{E}^*$. Then:*

$$\mathcal{E}^*(u) = \mathcal{E}^*(v) \implies u \overset{\circ}{=} v$$

The proof is derived in a simila manner to the Theorem 3.1.

*Proof.* Take a context-label pair $\rho, y$ sample with symbols $\sigma_1, \ldots, \sigma_N$, one of which will be $u$ and one of which will be $v$. Further, let $i_u$ and $i_v$ be te indices of $u$ and $v$ in the set of symbols. Then, by Lemma A.1, we know that:

$$p(u|\rho, y) = \exp(\text{sim}(\mathcal{E}^*(u), E^*(y, \rho)))p(u) \left( \frac{p(u|\rho, y)}{p(u)} + \frac{p(v|\rho, y)}{p(v)} + \underbrace{\sum_{\substack{j \\ j \neq i_u, i_v}} \frac{p(\sigma_j|\rho, y)}{p(\sigma_j)}}_{\alpha} \right)$$

By isolating the exponential term, we get:

$$\frac{p(u|\rho,y)}{\left(\frac{p(u|\rho,y)}{p(u)} + \frac{p(v|\rho,y)}{p(v)} + \alpha\right)p(u)} = \exp(\mathrm{sim}(\mathcal{E}^*(u), E^*(y,\rho)))$$

Similarly, for $v$, we have:

$$\frac{p(v|\rho,y)}{\left(\frac{p(u|\rho,y)}{p(v)} + \frac{p(v|\rho,y)}{p(v)} + \alpha\right)p(v)} = \exp(\mathrm{sim}(\mathcal{E}^*(v), E^*(y,\rho)))$$

By hypothesis, we know that $\mathcal{E}^*(u) = \mathcal{E}^*(v)$. Therefore, the exponentials terms are equal. This implies the following:

$$\frac{p(u|\rho,y)}{\left(\frac{p(u|\rho,y)}{p(u)} + \frac{p(v|\rho,y)}{p(v)} + \alpha\right)p(u)} = \frac{p(v|\rho,y)}{\left(\frac{p(u|\rho,y)}{p(v)} + \frac{p(v|\rho,y)}{p(v)} + \alpha\right)p(v)}$$

$$\frac{p(u|\rho,y)}{p(u)} = \frac{p(v|\rho,y)}{p(v)}$$

Note that, no assumption is made on $\rho$ or $y$. Therefore, using $p(\rho|u) = p(\rho|v)$ concludes the result ($u \overset{\circ}{=} v$).

$\square$

## A.5  EMBEDDING SYMBOL FUNCTION UPDATE RULE

Let us consider the infoNCE loss with the scalar product as similarity function:

$$\mathcal{L}(D,\theta) = -\sum_i \log\left(\frac{\exp\left(\mathcal{E}(\sigma_i;\theta)^T E(y_i,\rho_i;\theta)\right)}{\sum_j \exp(\mathcal{E}(\sigma_j;\theta)^T E(y_j,\rho_j;\theta))}\right)$$

Now, let us assume that $\mathcal{E}$ maps each symbol, $w_i$, to a corresponding parameter vector, denoted as $\overline{w}_i$. For simplicity, let us call $E(y_j,\rho_j;\theta) = \overline{c}_j$. Next, we differentiate the infoNCE loss wrt. $\overline{w}_i$.

$$\frac{\partial \mathcal{L}(D,\theta)}{\partial \overline{\sigma}_k} = \frac{\partial}{\partial \overline{\sigma}_k}\left(-\sum_i \log\left(\frac{e^{\overline{\sigma}_i^T \overline{c}_j}}{\sum_j e^{\overline{\sigma}_i^T \overline{c}_j}}\right)\right)$$

$$= -\sum_i \delta_{ik}\overline{c}_i - \frac{1}{\sum_j e^{\overline{\sigma}_j^T \overline{c}_j}}\sum_j e^{\overline{\sigma}_i^T \overline{c}_j}\delta_{ik}\overline{c}_j$$

$$= -\overline{c}_k + \sum_j \underbrace{\frac{e^{\overline{\sigma}_k^T \overline{c}_j}}{\sum_t e^{\overline{\sigma}_k^T \overline{c}_t}}}_{0 \le \alpha_j \le 1}\overline{c}_j = -\overline{c}_k + \sum_j \alpha_j\overline{c}_j$$

Thus, the update rule with learning rate $\eta$:

$$\overline{\sigma}_k \leftarrow \overline{\sigma}_k + \eta(\overline{c}_k - \sum_j \alpha_j c_j)$$

Since $\sum_j \alpha_j = 1$, the update on $\overline{\sigma}_k$ pushes towards $\overline{c}_k$ and away from the weighted sum of all $\overline{c}_j$.

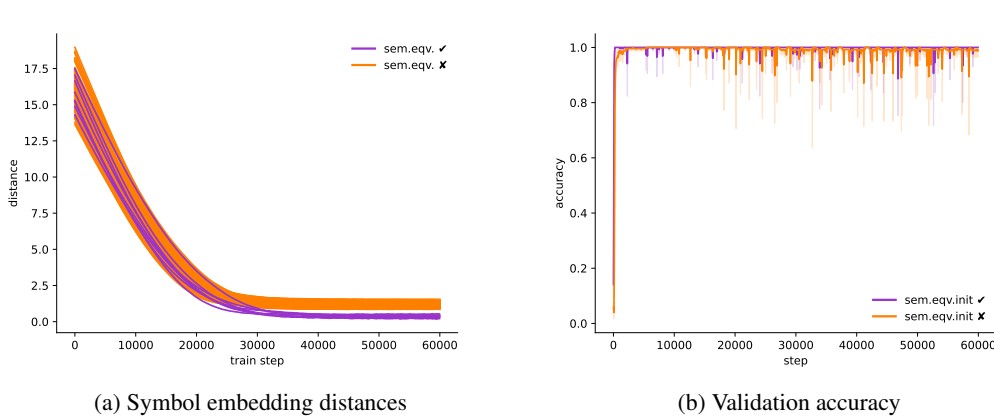

(a) Symbol embedding distances

(b) Validation accuracy

Figure 4: Embedding distances and validation accuracy for bigger architectures

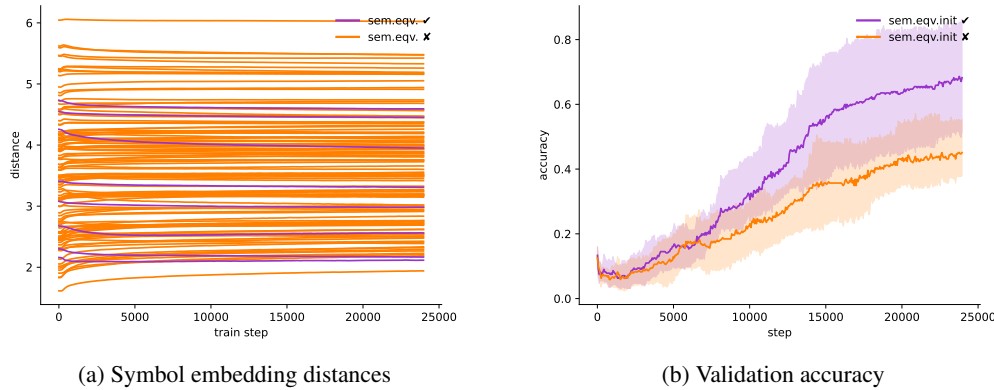

(a) Symbol embedding distances

(b) Validation accuracy

Figure 5: Embedding distances and validation accuracy for architecture without weight decay

## A.6 EXPERIMENTS

### A.6.1 BIGGER EMBEDDINGS

One of the hypotheses in Theorems 3.1 and 4.2 concerns the existence of context-label pairs that form a basis for $\mathbb{R}^d$, where $d$ is the embedding dimension. Consequently, when the embedding dimension is sufficiently large, there is no guarantee that the model will encode semantically equivalent symbols in close proximity. However, even when the embedding dimension is increased to 128 (which is 16 times larger than required by the theorems), the model still represents semantically equivalent symbols near each other (see Figure 4a).

In Figure 4b, we report the validation accuracy for a model initialized with pre-trained representations (shown in purple) and the same model with randomly initialized symbol representations (shown in orange). Both models perform equally well, suggesting that if the model's capacity is sufficiently large, it can effectively compensate for differences in initialization.

### A.7 NO WEIGHT DECAY

It is instructive to introduce a failure case. In this experiment, we disable weight decay during training, removing the regularization that encourages the model to use low-magnitude weights. The results are shown in Figure 5a. Without weight decay, the model fails to encode semantically equivalent symbols (shown in purple) close to each other within a reasonable time. However, a trend toward pushing semantically equivalent symbols closer together is still observable. This results in

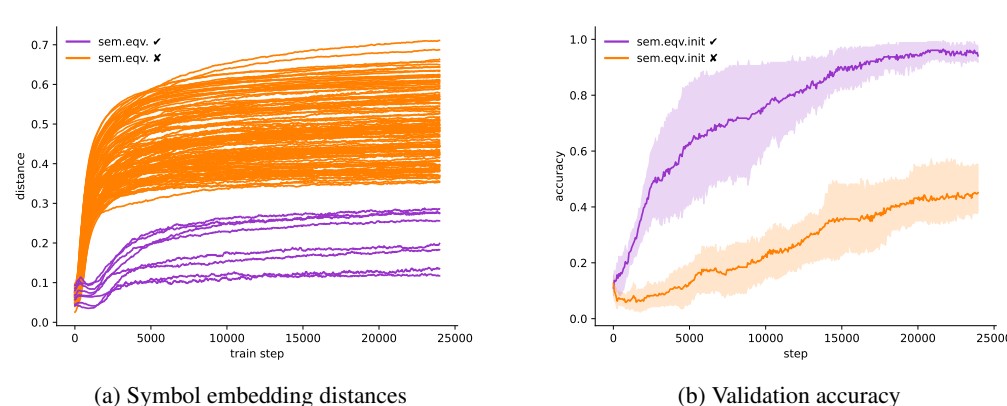

(a) Symbol embedding distances        (b) Validation accuracy

Figure 6: Embedding distances and validation accuracy for architecture without weight decay but with low embedding initialization

semantic equivalence relationships that are only marginally better than those produced by a randomly initialized model.

The effect of poor semantic encoding is visualized in Figure 5b. The model initialized with pre-trained embeddings (shown in purple) performs only slightly better than the randomly initialized model (shown in orange).

Furthermore, we speculate that weight decay may have additional effects on the model's capacity to encode semantic equivalence relationships. As weight decay pushes all embeddings toward zero, the model is discouraged from encoding semantically different symbols too close to each other, as doing so would eventually make it unable to distinguish between them. This limitation does not apply to semantically equivalent symbols; if they become indistinguishable, the model's performance would still be correct.

### A.8 No Weight Decay & Low Weight Initialization

The failure caused by the absence of weight decay can be mitigated by employing a lower initialization scheme. In this experiment, we initialize the embeddings using a normal distribution with a mean of $0$ and a standard deviation of $0.02$. The results are shown in Figure 6a. Under this scheme, the model is able to encode semantically equivalent symbols (shown in purple) close to each other within a reasonable time. Additionally, the model initialized with pre-trained embeddings (shown in purple) performs better than the randomly initialized model (shown in orange), as seen in Figure 6b.

Since all the embeddings are initialized with low-magnitude values, the distance between symbols starts small, compared to the standard PyTorch initialization, which uses a normal distribution with a mean of $0$ and a standard deviation of $1$ [1], as shown in Figure 2a.

### A.9 Weight Decay & Sligthly Low Weight Initialization

Another scenario worth visualizing is the case where we employ both weight decay and low-weight initialization. The results are shown in Figure 7a. In this setup, the model successfully encodes semantically equivalent symbols (shown in purple) close to each other within a reasonable time. Consequently, the model initialized with pre-trained embeddings (shown in purple) outperforms the randomly initialized model (shown in orange), as seen in Figure 7b.

Interestingly, despite the presence of weight decay, the distances between semantically different embeddings end up being larger compared to the previous experiment, which did not use weight

---

[1] https://github.com/pytorch/pytorch/blob/v2.4.0/torch/nn/modules/sparse.py#L14

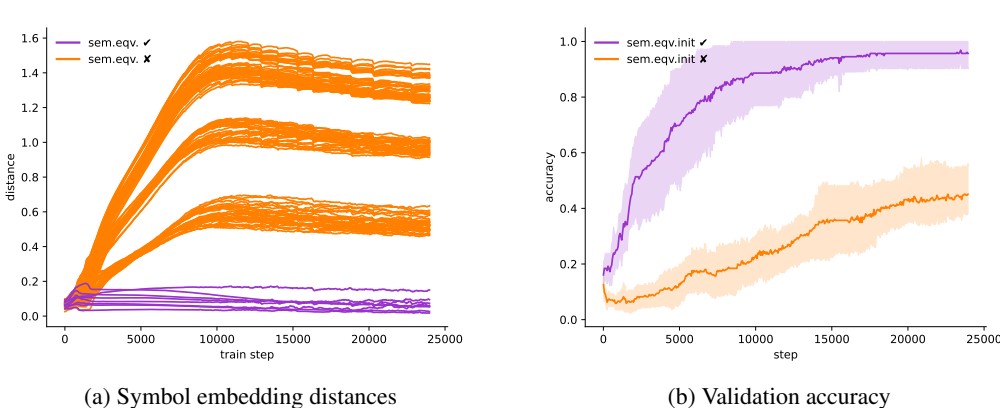

(a) Symbol embedding distances

(b) Validation accuracy

Figure 7: Embedding distances and validation accuracy for architecture with weight decay but with low embedding initialization

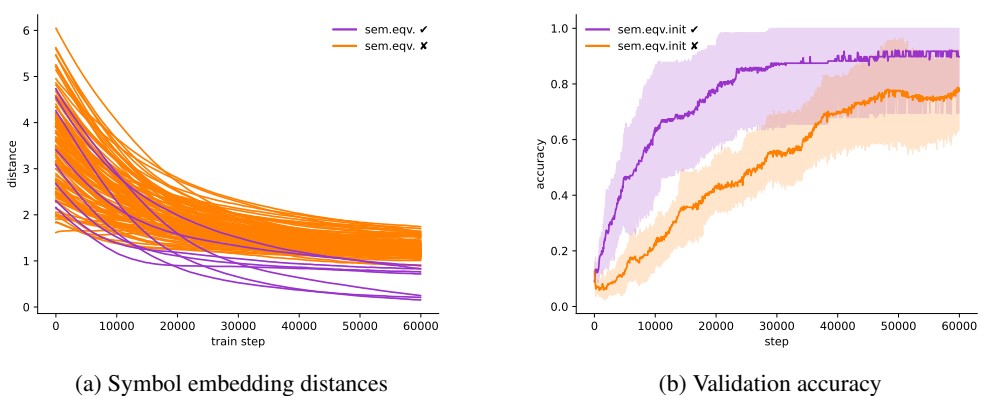

(a) Symbol embedding distances

(b) Validation accuracy

Figure 8: Embedding distances and validation accuracy for a slightly misaligned contrastive and classification datasets

decay. Additionally, the distances between semantically different embeddings appear divided into three distinct bands. Unfortunately, we do not yet have a clear explanation for this behavior.

## A.10 SLIGTHLY MISALIGNED CONTRASTIVE DATASET

The proposed theory suggests that when a contrastive dataset and a classification dataset are distributionally aligned (see Definition 4.1), training on the contrastive dataset should produce embeddings that are semantically meaningful for the target classification task. But what happens if the distributional alignment is imperfect? In this experiment, we introduce a slight misalignment between the contrastive and classification datasets.

The misalignment is created by assigning a fixed probability to all correct contexts of a particular symbol. Specifically, we assign a probability of $0.9$ to one chosen context, while the remaining probability is evenly distributed among all other contexts.

The results are presented in Figure 8a and 8b. The model is still able to position semantically equivalent symbols (shown in purple) close to each other in a reasonable amount of time. Furthermore, the model initialized with pre-trained embeddings (shown in purple) outperforms the randomly initialized model (shown in orange), as seen in Figure 8b. However, compared to the perfectly aligned case (see Figures 2a and 2b), the model takes longer to bring semantically equivalent symbols closer together. As a result, the generated embeddings are less effective for the classification task.

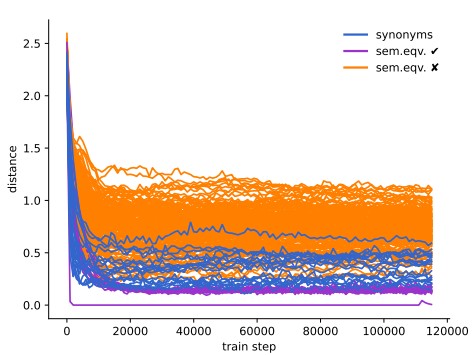 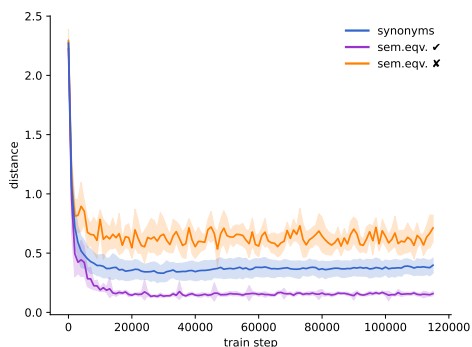

(a) Symbol embedding distances between semantically different words (orange), naturally semantically similar (blue), artificially semantically equivalent (purple)

(b) Symbol embedding distances averaged between semantically different words (orange), naturally semantically similar (blue), artificially semantically equivalent (purple)

Figure 9: Embedding distances for the Word2Vec task. The train steps corresponds to $1\%$ of the Bookcorpus dataset.

### A.11    WORD EMBEDDINGS

Until now, we have exclusively experimented with algorithmic data. However, real-world data are far from being as clean as the data we have used so far. In this experiment, we aim to train a Word2Vec-like model Mikolov et al. (2013) on a small text corpus. Specifically, we will use the BookCorpus dataset Zhu et al. (2015) as our source. To access this dataset[2], we will utilize Hugging Face's datasets library Lhoest et al. (2021). Our objective is to observe whether the patterns identified in algorithmic data are replicated in this real-world context.

We will also artificially inject semantic equivalences into the dataset by replacing certain words with fabricated perfect synonyms. For instance, a word such as good may be replaced with equal probability by $good_1$ or $good_2$ within the dataset. Specifically, we selected 100 words and replaced each with one of the fabricated synonyms. Further, we will compare these artificially created synonyms with naturally occurring synonyms.

To generate symbol-context pairs, we will sample a sentence and randomly mask one word. The masked word will serve as the symbol, while the remainder of the sentence will act as the context.

We will train a Transformer encoder-only model[3] to generate context-embedding. The symbol embedding will be generated by a simple embedding layer. [4] As previously, we will use the InfoNCE loss van den Oord et al. (2018) to match the context-embedding with the correct symbol-embedding, while pushing away the incorrect symbol-embeddings.

During training, we expect the embeddings of semantically similar symbols to converge, while embeddings of unrelated symbols should remain far apart. This behavior is confirmed in Figure 9. Specifically, we observe that the embeddings of semantically equivalent symbols (shown in purple) converge to a small distance, whereas the embeddings of semantically different symbols (shown in orange) remain distant. In blue, we have naturally occuring synonyms which position themeselves in between the perfect semantically equivalent and the semantically different symbols.

---

[2]https://huggingface.co/datasets/bookcorpus/bookcorpus

[3]https://pytorch.org/docs/stable/generated/torch.nn.Transformer.html

[4]https://pytorch.org/docs/stable/generated/torch.nn.Embedding.html

