# OpenReview forum: "Contrastive Learners Are Semantic Learners"
_ICLR.cc/2025/Conference — Submitted to ICLR 2025_

### Official Review · Reviewer_5kK9 · 2024-11-03

**Soundness:** 2
**Presentation:** 3
**Contribution:** 2
**Rating:** 5
**Confidence:** 4

**Summary:**

This work presents an analysis that contrastive loss (InfoNCE) learns representations that encode semantic relationships effective for downstream tasks. The paper proves that, under certain assumptions, semantically equivalent symbols have the same learned embeddings.

**Strengths:**

1. The paper borrows frameworks from programming languages to analyze theoretical properties of contrastive embeddings, which is an original and interesting perspective.
2. The paper is written clearly. The exposition is very good for building up reader intuition and the assumptions are clearly stated for each proof.

**Weaknesses:**

> Weakness 1. Corollary 4.3 states an **iff** relationship between the alignment of symbols and their respective embeddings. However, the authors only prove the forward direction.

Conditional on Corollary 4.3 being a typo, and only the forward direction holding: $u \doteq_{\mathcal{D}} v \Rightarrow \mathcal{E}^*(u) = \mathcal{E}^*(v)$, why is this conclusion valuable? An encoder that maps every input to 0 also has this property.

> Weakness 2. The assumptions are unrealistic and lead trivially to the Theorems 3.1 and 4.2.

I believe the proof for Theorem 3.1 is a bit obfuscated. The authors essentially make two assumptions:

1. that contrastive encoders learn the following probability ratio up to a multiplicative constant: $f_i = c \cdot \frac{p(\rho, y \mid \sigma_i)}{p(\rho, y)}$. (taken from van den Oord et al. (2018))
2. that $p(\rho, y \mid u) = p(\rho, y \mid v)$, which is a trivial result of (1) $u \doteq_{\mathcal{D}} v$, which implies to $p(y \mid u, \rho) = p(y \mid v, \rho)$,  and (2) $p(\rho \mid u) = p(\rho \mid v)$.

Because of Assumption 2, we can immediately conclude that $f_u = f_v$, and use the basis argument to conclude that $u$ and $v$ must have the same embeddings.

The direct assumption that $p(\rho, y \mid u) = p(\rho, y \mid v)$ seems quite strong and substantially simplifies the analysis, as the desired result follows almost immediately from this condition. Also, it is unclear that this result is meaningful in any way for learnability. As stated previously, the encoder that maps every input to 0 also satisfies this property. It is only this property in conjunction with the reverse direction *(that semantically dis-similar symbols are not mapped to the same encoding)* that is interesting. I may be misunderstanding, but I do not see any proof of the reverse direction in the paper.

Theorem 4.2 follows the same basic structure, and the same concerns apply.

 > Weakness 3. Limited experimentation

The ModAdd experiment, while very illustrative of the beneficial properties of contrastive learning, can be solved symbolically. It lacks complexity and noise seen in real data. Simple experiments in language or vision would better illustrate the developed theoretical results. Possibly an evaluation on an MLM task (as shown as an example in Section 2.1)?

---
References:

Aaron van den Oord, Yazhe Li, and Oriol Vinyals. Representation Learning with Contrastive Predictive Coding. 2018.

**Questions:**

Mentioned in weaknesses.

Some additional remarks:
1. In Figure 2(a), the semantically equivalent pairs on average converge to a lower euclidean distance compared to non-semantically equivalent pairs. However, it does not converge to 0, as the previously developed theory would suggest. Why is this the case?

2. Minor remark: There should be better ways of motivating the assumption $E^*(\rho_i, y_i)$ forms a basis for $\mathbb{R}^d$. There exists prior work on hyper-spherical contrastive learning that proves certain uniformity results that seem related to the author's assumption of uniformity [1].

*Minor*
- line 176: mentions "both" but only lists Theorem 3.1.
- line 385: "potetial"
- Appendix A.9 Title: "SLIGTHLY"

---
References:

[1] Understanding Contrastive Representation Learning through Alignment and Uniformity on the Hypersphere. Tongzhou Wang, Phillip Isola 2022

---

> ### Author Response · Authors · 2024-11-13
>
> Thank you for carefully reviewing the mathematical aspects of our paper. We hope that the following considerations may lead you to reconsider your evaluation of the paper.
>
> ---
> ### Weakness 1
>
> We confirm that Cor. 4.3 indeed represents an iff relationship. We agree that the backward direction was not explicitly demonstrated. The proof is simple, and we will add a full derivation in the next revision. In the meantime, here is the outline:
>
> - As you mentioned, from [2] we have $f_i = c \frac{p(\rho, y | \sigma_i)}{p(\rho, y)}$ or, using our notation, $\mathcal{E}^*(u) = c \frac{p(\rho, y | u)}{p(\rho, y)}$.
> - A similar relationship holds for $v$, so $\mathcal{E}^*(v) = c \frac{p(\rho, y | v)}{p(\rho, y)}$.
> - By hypothesis, $\mathcal{E}^*(u) = \mathcal{E}^*(v)$, which implies $p(\rho, y | u) = p(\rho, y | v)$.
> - Using $p(\rho | u) = p(\rho | v)$, we get $p(y | \rho, u) = p(y | \rho, v)$, fulfilling $u \circeq v$.
>
> If the constant $c$ raises concerns, the same steps apply from the end of Lemma A.1 as in Th. 3.1.
>
> ### Weakness 2
> > Assumption $f_i=c\frac{p(\rho,y|\sigma_i)}{p(\rho,y)}$.
>
> This assumption arises from a model optimality condition rather than being directly borrowed.
>
> > Assumption $p(\rho,y|u)=p(\rho,y|v)$.
>
> We agree with the reviewer that the assumption $p(\rho,y|u)=p(\rho,y|v)$ is quite strong. However, we want to stress that this assumption stems from the definition of semantic equivalence ($p(y|u,\rho)=p(y|v,\rho)$) and the assumption $p(\rho|u)=p(\rho|v)$. While the former is the relationship under study, the latter is introduced to make the problem tractable. However, we want to emphasize that such assumption is quite reasonable even in practice. If it were not to hold, we could find ourselves in a scenario where a context always appears together with one symbol but never with the other. In such cases, it would be impossible to recognize the two symbols as semantically equivalent.
>
> > The encoder that maps every input to 0 also satisfies this semantical equivalence property.
>
> The forward implication of Cor. 4.2 is significant, even without the backward implication. While a zero-encoder satisfies the forward implication, it is unlikely to be optimal except in degenerate data scenarios. We find it compelling that, if two symbols are semantically equivalent in classification data, the optimal encoder for contrastive data (under Def.4.1) must reflect this equivalence in its embeddings, facilitating easier training on the classification data.
>
> We will address these points in the manuscript.
>
> ### Weakness 3
> > The ModAdd experiment lacks complexity and noise seen in real data.
>
> Our goal is to provide a theoretical explanation for how semantically similar objects are encoded with similar embeddings using contrastive learning and joint embedding methods, a concept well-supported in the literature. As such, we limited our experiments to a simplified example, but we agree that a larger-scale experiment would enhance the study.
>
> We propose the following experiment inspired by word2vec:
>
> - Use SimCLR to learn word representations from natural text.
> - Randomly select $k$ words and generate semantically equivalent variants for each.
> - Randomly swap occurrences of the original word with its variant (e.g., swapping "cat" with a new token, $\text{cat}_1$ and $\text{cat}_2$, with probability $p = 0.5$).
>
> Since these tokens are semantically equivalent, we expect the model to bring their embeddings closer. We can also analyze how varying $p$ affects the model's ability to learn this relationship.
>
> Please let us know if this experiment would better address your concerns. In this case, we will include it in the revised manuscript.
>
> ### Remark 1
> > In Figure 2(a), the semantically equivalent pairs do not converge to 0. Why?
>
> At a certain point during training, the contrastive loss becomes so small that further changes in the embeddings are minimal, even after 10,000 epochs.
>
> We will include this explanation in the revised manuscript.
>
> ### Remark 2
> > Basis assumption motivation.
>
> The assumption of a basis is made so that the only possible solution for the optimal model is to encode semantically equivalent symbols in the same embedding. Furthermore, note that any slight perturbation of $E^*(\rho_{i_0},y_{i_0}), \dots, E^*(\rho_{i_d},y_{i_d})$ would establish a basis almost surely. This hypothesis is also employed in the context of invertible neural networks [1].
>
> However, we recognize the mentioned manuscript as a missed related work that will be included in the next revision.
> ### Minors
> > - line 176: mentions "both" but only lists Theorem 3.1.
> > - line 385: "potetial"
> > - Appendix A.9 Title: "SLIGTHLY"
>
> Thank you for pointing out these typos. We will correct them in the revised manuscript.
>
> [1] Finzi, Marc, et al. Invertible convolutional networks. 2019.
>
> [2] Aaron van den Oord, et al. Representation Learning with Contrastive Predictive Coding. 2018.

---

> ### Comment · Reviewer_5kK9 · 2024-11-15
> **Response**
>
> Thank you for your response! The outline of the theorem's reverse direction looks correct to me. The proposed new experiment would also better address my concerns. If the authors revise the paper to include the reverse direction of Corollary 4.3, and add the new proposed experiment to the paper, I will raise my score.
>
>  I am still concerned about the strength of the assumptions made in the paper. Is there anything that can be said about the "closeness" of embeddings with very similar (but not equal) conditional distributions?

---

> > ### Author Response · Authors · 2024-11-16
> >
> > We have revised and uploaded the new version of the manuscript. A complete change log is provided in the following comment to this one.
> >
> > > I am still concerned about the strength of the assumptions made in the paper. Is there anything that can be said about the "closeness" of embeddings with very similar (but not equal) conditional distributions?
> >
> > We appreciate this concern and would like to provide a detailed clarification.
> >
> > Since contrastive learning operates without access to labels, no formal semantics can be directly encoded in the generated representations. The training process is guided solely by the conditional distribution of context-symbol pairs. Specifically, if a symbol $u$ frequently co-occurs with a context $\rho$, their representations will be positioned close to each other in the embedding space.
> >
> > Consider a third symbol $v$ that shares the same conditional distributions as $u$. In this case, the representation of $v$ must position itself to behave similarly to the representation of $u$ with respect to the contexts. Assuming the contexts form a basis, the only possible solution for $u$ and $v$ to share behavior is to share representations (as demonstrated in Theorem 4.2).
> >
> > Let us now examine the case where $v$ and $u$ have antipodal conditional distributions with respect to the contexts. This means that $v$ tends to be close to certain contexts while $u$ tends to be far from them (and vice versa). In this situation, it is impossible for $u$ and $v$ to share representations in the optimal contrastive model, as this would require them to behave similarly, which contradicts their antipodal distributions.
> >
> > For the case where $v$ and $u$ have similar but not equal conditional distributions, some contexts appear frequently with $u$ but less frequently with $v$ (and vice versa). Intuitively, this scenario can be addressed by positioning the representations of $u$ and $v$ close to, but not equal to, each other. While this is not guaranteed to be the unique solution—there may exist solutions where $\mathcal{E}(u)$ and $\mathcal{E}(v)$ are far apart while maintaining similar behaviors with respect to the context—we hypothesize that under appropriate conditions (similar to assuming contexts form a basis), the optimal model will position $\mathcal{E}(u)$ and $\mathcal{E}(v)$ close together.
> >
> > Our empirical evidence in Section A.10 of the appendix supports this intuition. We conducted experiments where we deliberately disrupted conditional equivalences for semantically equivalent symbols. Figure 8.a shows that the separation between semantically equivalent and different symbols begins to blur. This is because the contrastive learning procedure does not actually care of what we believe the labels to be, but it cares only to make context-symbol pairs that appear often close, and context-symbol pair that do not appear far apart.
> >
> > While the assumption $p(\rho|u)=p(\rho|v)$ remains idealized in typical contrastive learning scenarios, we believe that Theorem 3.1, combined with these insights, provides evidence suggesting an isometry between conditional distributions and the Euclidean distance of the optimal representation (under certain assumptions about the contexts, similar to the basis assumption). While it would definitely be interesting to show that such an isometry exists. We believe that this statement falls outside of the scope of the current work as we focus on the conditions under which formal semantics can be encoded by a contrastive algorithm.
> >
> > We trust this detailed explanation addresses your concerns about our assumptions.

---

> > > ### Author Response · Authors · 2024-11-16
> > > **Change Log**
> > >
> > > ### Main changes
> > > - Added a backward proof for Corollary 4.3 in Appendix Section A.4.
> > > - Clarified the assumption $p(\rho|u) = p(\rho|v)$ in the "Threats to Validity" section (Section 7).
> > > - Addressed assumption $p(\rho|u) = p(\rho|v)$ in Section 3 and 4.
> > > - Addressed the issue of embedding distances not being zero (Section 5).
> > > - Addressed alignment hypothesis in the "Threats to Validity" section (Section 7).
> > > - Provided a stronger motivation for the basis hypothesis (Section 3).
> > > - Corrected minor typographical errors.
> > > - Added a Word2Vec-style experiment in Appendix Section A.11.
> > > - Improved terminology by revising the use of "semantic relations" (often in the context of "semantic equivalence relations") and "semantic learning" to avoid misinterpretation related to linguistic semantics.
> > > - Revised the abstract, introduction, discussion, conclusion, and threats to validity sections to better clarify the scope and contributions of the work.
> > > - Specified "semantic relations" in RQ2.
> > > - Provided a more precise answer to RQ2.
> > > - Added a "Future Work" section (Section 10), which includes:
> > >   - Exploring multitask learning parallelism.
> > >   - Considering the weakening of assumptions.
> > > - Included a cat-vs-dog example in Section 2.1.
> > >
> > > ### Added References
> > >
> > > - Tongzhou Wang and Phillip Isola. *Understanding contrastive representation learning through alignment and uniformity on the hypersphere.* In *International Conference on Machine Learning*, pp. 9929–9939, PMLR, 2020.
> > > - Levy, Omer, and Yoav Goldberg. *Neural word embedding as implicit matrix factorization.* Advances in Neural Information Processing Systems 27 (2014).
> > > - Andreas Maurer, Massimiliano Pontil, and Bernardino Romera-Paredes. *The benefit of multitask representation learning.* Journal of Machine Learning Research, 17(81):1–32, 2016.
> > >
> > > ### Awaiting response on
> > > - Experiment with alternative contrastive architecure.
> > > - Experiment with triplet loss instead of InfoNCE loss.

---

> > > ### Comment · Reviewer_5kK9 · 2024-11-26
> > > **Reviewer Response**
> > >
> > > Thank you for adding the reverse direction of the proof and providing a detailed change log. Given the new proof and the additional Word2Vec experiment, I will increase my score.
> > >
> > > However, I believe the Word2Vec experiment to be too synthetic. Using artificial synonyms seems counter-intuitive when we really want to see if actual synonyms are embedded close to each other. Is it true that "real" synonyms are equivalent/close to each other in latent space?

---

> > > > ### Author Response · Authors · 2024-11-26
> > > >
> > > > We updated the manuscript with a figure showcasing the distance between a list of $20$ semantically similar words.The distances between these naturally occuring synonyms are positioned between the semantically different ones and semantically equivalent ones, as one would expect. We appreciate the reviewer’s feedback and hope that the following considerations might lead you to reconsider your evaluation of our paper.
> > > >
> > > > However, we want to point out what it appears to be a misunderstanding. Please let us know if the following explanation is clear.
> > > >
> > > > > However, I believe the Word2Vec experiment to be too synthetic. Using artificial synonyms seems counter-intuitive when we really want to see if actual synonyms are embedded close to each other. Is it true that "real" synonyms are equivalent/close to each other in latent space?
> > > >
> > > > 1. Firstly, we will need to formalize the term **synonym**. Formally, we will say that two words are synonyms if they are semantically equivalent. Meaning that one can be used in place of the other without affecting the label distribution of the target task. This is also the definition of _semantical equivalence_ used in the paper.
> > > >
> > > > Unless specified otherwise, in these notes, we speak of synonyms with their formal connotation. However, a comparison with the informal connotation will be necessary.
> > > >
> > > > This definition of synonymy does have very important implications. The most apparent is its dependence on the target task. For example, "grass" and "sky" are not synonyms for a masked language modeling task. However, for a task where we require to output the number of tokens in a sentence, they are synonyms (as replacing one with the other does not change the number of words in the sentence).
> > > >
> > > > 2. It is desirable to have synonyms equal to each other in the embedding space. So that swapping one with the other does not change the output distribution of the network by definition. This is the same principle that one uses when designing, for example, permutation-invariant networks.
> > > >
> > > > 3. When we train a contrastive learning algorithm, we do not capture the downstream-task semantics because we do not have access to the labels (generally speaking). However, we do capture the conditional distribution of the words. Meaning that if two words appear in the same context (with similar frequency), then they will be close in the embedding space.
> > > >
> > > > 4. Ok, but why synonyms (here intended with their linguistic connotation) are encoded close to each other? This is because, in natural language, if two words can appear in the same context, they are likely to be semantically similar (here semantically is intended with its linguistic connotation). This is often referred to as a "Distributional Hypothesis" which is used in algorithms such as word2vec.
> > > >
> > > > 5. Ultimately, it is only thanks to the distributional property of natural language that we are seeing semantically similar (linguistic connotation) words close to each other in the embedding space. One should not expect to see the same results when this property is not satisfied.
> > > >
> > > > 6. The last step one has to do to accept this work, is to accept the formal definition (Def. 2.1) of semantic equivalence as the formalization of its linguistic counterpart, which we argue to be well-grounded.

---

### Official Review · Reviewer_t6m4 · 2024-11-03

**Soundness:** 2
**Presentation:** 2
**Contribution:** 1
**Rating:** 5
**Confidence:** 4

**Summary:**

The paper explores contrastive learning and investigates under which conditions contrastive learning can be effective for downstream tasks. The paper introduced a "distribution alignment hypothesis": if the data distributions used in contrastive learning and downstream tasks are aligned, then contrastive learning will learn semantic representations that good for the task. To support the hypothesis, the paper provides a controlled experiment using a mod-addition task.

**Strengths:**

At a high level, the work is well motivated: the match between the data used in contrastive learning and final downstream task is crucial for good empirical performance.

**Weaknesses:**

- The main limitation I see is similar to that the authors identify in Section 7: the alignment hypothesis introduced is very strong and verifying it is not possible in practice. On the other hand, the intuitions that it sheds light on are not novel: despite the claims made, it's been known for a long time that one can learn similarity from this type of data (please see work such word2vec, or subsequent papers such as Levy and Goldberg, showing the relation to the classic distributional semantics work)
- The paper considers a very specific form of contrastive learning using a specific augmentation function. In general the term contrastive learning is much wider, and the data used varies widely, from data with labels in classification, to question answer pairs in retrieval. I would recommend being much more specific in the claims made.
- The paper is very loose with the terminology that is at the basis at its main questions and claims, leading to vague or in-accurate statements. For example the answer to RQ2 in intro is clearly yes: we can train embeddings to reflect semantic similarity (see more in suggestions below). Similarly, what is a semantic learner? Semantics is a very wide term (see the field of semantics in linguistics) and yes, speaking in general terms, we already know we can learn meaning from raw data.

**Questions:**

The paper needs to provide definitions or be very specific about the terms such as "semantic relations". For example in RQ2 (line 47): "can we train embeddings that effectively encode semantical relations?" What do you mean by semantic relations here? And the answer to this question is yes, clearly more than ten years of research in the field of distributional semantics have shown that we can train them to reflect semantic similarity.
I would also encourage the authors to try and get to core of their findings and explain them better: I struggled to figure out if there is actual content to the definitions and theorem given, or I was simply walked through re-writing of formulas. For this, the paper would benefit from focusing on the actual setup (for example the SimCLR variants used) and less on making wider claims about the findings.

---

> ### Author Response · Authors · 2024-11-14
>
> Thanks for your feedback. We believe that many of the concerns arise from overloading the term semantic with both a linguistic and a technical meaning. In the following revision, we will clarify these issues. We hope that the following considerations may lead you to reconsider your evaluation of the paper.
>
> ### Weakness 1
>
> > The alignment hypothesis (AH) is too strong.
>
> We agree. However, we would like to point out that, in practice, the conditions under which pretraining data can be useful for a downstream task are also quite strong. For instance, randomly generated data is unlikely to benefit any downstream task. Similarly, using genomic data as contrastive data would likely be ineffective for a sentiment classification task.
>
> > The AH is not useful in practice.
>
> Verifying AH in practice is not feasible. However, it can explain why some contrastive data are more effective than others for a particular downstream task.
>
> > Not novel intuitions
>
> We respectfully disagree with the reviewer’s assessment. While the intuition that contrastive learning can capture semantics (here, as a linguistic term) is well established, our work aims to formalize the conditions under which this occurs for the formal definition of semantics. We believe this adds a novel contribution to the field.
>
> ### Weakness 2
>
> > Too broad claims
>
> We agree, and we appreciate this feedback. We acknowledge that the title may have contributed to this impression, and we are considering alternatives such as "When Can a Contrastive Learner Capture Semantics?". Would this title be more appropriate?
>
> We will revise the claims in the abstract, introduction, and discussion to more accurately reflect the theoretical findings.
>
> ### Weakness 3
>
> > Loose terminology.
>
> We apologize for any confusion caused by the terminology. Our concept of "semantics" is based on Def.2.1, which is derived from classical formal semantics. We briefly reference the denotational notation [3]. In short, we define $u$ and $v$ as **semantically equivalent/similar** if they can be interchanged in any/most contexts $\rho$ without affecting the outcome distribution $y$. We recognize that using "semantics" as both a linguistic and technical term may have led to confusion.
>
> > Unclear meaning of "semantic learner"
>
> We apologize for the confusion. By "semantic learner," we refer to a model that encodes semantic equivalence relationships (as defined in Def.2.1) within its embeddings.
>
> > We already know that contrastive learning (CL) can learn semantics from the literature.
>
> We agree that it is well established in the literature that CL can capture semantics (here intended as a linguistic term). However, our work aims to identify and formalize the theoretical conditions under which this occurs for the formal definition of semantics. We believe this adds a novel contribution to the field.
>
> > Unclear meaning of "semantic relations"
>
> We apologize for the confusion. Here, "semantic relations" specifically refers to the semantic equivalence relations in Def.2.1.
>
> > Not novel RQ2.
>
> We agree that the question "Can we train embeddings that effectively encode semantic relations?" may appear unoriginal if interpreted with the linguistic meaning of semantics. However, in our work, "semantic relations" refers to formal semantic equivalence, as in Def.2.1.
>
> We will address all the previous terminology issues in the revised manuscript.
>
> ### Weakness 4
>
> > Findings not clearly stated / too broad claims.
>
> We apologize for the lack of clarity. We will revise the *Findings* paragraph in Section 1 to more clearly state the contributions of the paper and avoid overly broad claims.
>
> Additionally, we will introduce a paragraph at the end of the discussion to address the implications of Corollary 4.3. In the meantime, here is a brief explanation:
>
> - Suppose you have a cheap CL dataset, $\mathcal{C}$, which could be a set of images.
> - Suppose you have an expensive downstream dataset, $\mathcal{D}$, which could be a set of image-label pairs.
> - Suppose we use a CL method as described in Sect.3.
>
> We aim to learn useful embeddings from $\mathcal{C}$ to be used for learning $\mathcal{D}$. Formally, we desire
>
> $$u \circeq v \iff \mathcal{E}(u) = \mathcal{E}(v) \text{ ( denoted as ★)}$$
>
> What conditions on $\mathcal{C}$ and $\mathcal{D}$ are required to achieve ★? For once, we require an AH (Def.4.1) on the data. Further, when the AH holds, we can say that an optimal model must encode semantic equivalence relations in its embeddings, which characterizes a key property of the optimal model.
>
>
> [1] Goldberg, et al. Neural word embedding as implicit matrix factorization. 2014.
>
> [2] Arora, et al. A theoretical analysis of contrastive unsupervised representation learning. 2019.
>
> [3] Tennent et al. The denotational semantics of programming languages. 1976.

---

> > ### Author Response · Authors · 2024-11-16
> > **Change Log**
> >
> > We have updated the manuscript to include modifications requested by the reviewers. Below is a log of the changes made. We hope these revisions have enhanced the quality and clarity of the manuscript.
> >
> > ### Main Changes
> > - Added a backward proof for Corollary 4.3 in Appendix Section A.4.
> > - Clarified the assumption $p(\rho|u) = p(\rho|v)$ in the "Threats to Validity" section (Section 7).
> > - Addressed assumption $p(\rho|u) = p(\rho|v)$ in Section 3 and 4.
> > - Addressed the issue of embedding distances not being zero (Section 5).
> > - Addressed alignment hypothesis in the "Threats to Validity" section (Section 7).
> > - Provided a stronger motivation for the basis hypothesis (Section 3).
> > - Corrected minor typographical errors.
> > - Added a Word2Vec-style experiment in Appendix Section A.11.
> > - Improved terminology by revising the use of "semantic relations" (often in the context of "semantic equivalence relations") and "semantic learning" to avoid misinterpretation related to linguistic semantics.
> > - Revised the abstract, introduction, discussion, conclusion, and threats to validity sections to better clarify the scope and contributions of the work.
> > - Specified "semantic relations" in RQ2.
> > - Provided a more precise answer to RQ2.
> > - Added a "Future Work" section (Section 10), which includes:
> >   - Exploring multitask learning parallelism.
> >   - Considering the weakening of assumptions.
> > - Included a cat-vs-dog example in Section 2.1.
> >
> > ### Added References
> >
> > - Tongzhou Wang and Phillip Isola. *Understanding contrastive representation learning through alignment and uniformity on the hypersphere.* In *International Conference on Machine Learning*, pp. 9929–9939, PMLR, 2020.
> > - Levy, Omer, and Yoav Goldberg. *Neural word embedding as implicit matrix factorization.* Advances in Neural Information Processing Systems 27 (2014).
> > - Andreas Maurer, Massimiliano Pontil, and Bernardino Romera-Paredes. *The benefit of multitask representation learning.* Journal of Machine Learning Research, 17(81):1–32, 2016.
> >
> > ### Awaiting response on
> > - Experiment with alternative contrastive architecure.
> > - Experiment with triplet loss instead of InfoNCE loss.

---

### Official Review · Reviewer_iHA7 · 2024-11-04

**Soundness:** 3
**Presentation:** 4
**Contribution:** 3
**Rating:** 6
**Confidence:** 3

**Summary:**

The core argument is that if a pretraining task is "distributionally aligned" with a downstream task, this alignment benefits the downstream task. The concept of distributional alignment relies on "semantic equivalence" -- where 2 features are interchangeable for a prediction task without altering the target label's probability (e.g., synonyms). Two tasks are then distributionally aligned if they share these equivalences; that is, if tokens are semantically equivalent in one task, they remain so in another. The paper suggests that pretraining with a contrastive loss facilitates this by encouraging the model to capture these contextual equivalences on  context prediction task, making it beneficial for downstream tasks (when this alignment holds).

**Strengths:**

The paper presents an interesting theoretical exploration, backed by set of empirical experiments that, while somewhat limited, suppor t and illustrate the main theoretical arguments. Overall, the paper is clearly written, and the theoretical results, though potentially unsurprising, appear to offer novel insights.

**Weaknesses:**

Comments

1. This framework seems to have parallels with the learning theory in multitask learning. For instance, Maurer et al. (2016) (multitask subspace learning) argue that multitask learning can be advantageous when task-specific functions can be decomposed into shared and unique components, i.e., f_k = g_k cdot h, where function h is shared across tasks. Drawing on these connections with LT for MTL may be beneifical, and may clarify the novelty. Currently, previous research on MTL is not discussed in the paper.

2. The focus on token representations (word embeddings) as the primary benefit of pretraining is interesting, but it raises a question: could this analysis extend beyond token-level representations to contextualized models as a whole? It would be valuable to explore if and how these insights might generalize.

3. Lastly, the assumption of strict equivalence across all predictions might be too strong. A more nuanced setting, where only some predictions share semantic equivalences with the downstream task, would broaden the applicability. For example, if predicting continuations like "thumbs_up" and "thumbs_down" shares equivalences with a downstream sentiment classification task, but other alternative may not share them, the theory should ideally capture this partial alignment.  The intuition is that probably some of the pretrainintg decision share equivalencess, some unrelated, some require more 'fine-grain' equivalences' or more coarse-ones, this still works.

Reference: Maurer, A., Pontil, M, Romero-Paredes, B. (2016). The Benefit of Multitask Representation Learning. JMLR. https://jmlr.org/papers/volume17/15-242/15-242.pdf

**Questions:**

See the three comments above, I'd appreciate authors view on these points.

---

> ### Author Response · Authors · 2024-11-14
>
> We appreciate the reviewer's feedback highlighting important extensions and connections to related fields. We hope that the following considerations may enhance the paper's quality.
>
> ### Weakness 1
>
> > Parallels with the MultiTask Learning (MTL) literature.
>
> We agree that there is a parallelism between our work and the suggested field, specifically the work of [1]. The shared representation function $h$ is required to encode the information that is relevant to all tasks. Similarly, here, we want an embedding function that encodes the relevant information for the downstream tasks.
>
> The main difference lies in the fact that, during the downstream training, we allow for the embedding (or representation) function $\mathcal{E}$ (or $h$) to specialize for each downstream task. However, if we were to freeze its parameters, the methodology would reduce to learn a single $h$ for all $g_k$, similarly to [1].
>
> We speculate that the optimal $h^{*}$ should encode semantically equivalent relationships for the target tasks (under the right hypotheses).
>
> We will make sure to address this parallelism in the following revision.
>
> ### Weakness 2
>
> > The work is limited to token representation.
>
> We want to stress that the embedding function $\mathcal{E}$ (mapping symbols to symbol-embedding) and the encoder function $E$ mapping context to context-embeddings are not subject to restriction in either data modality nor architecture.
>
> For example, we may have an embedding function, implemented as a Convolutional Neural Network, generating symbol-embedding from a single image-patch. Meanwhile, we may have an encoder function, implemented as Vision Transformer, generating the context-embedding from the remaining image-patches.
>
> Similarly, we could have an embedding function, implemented as a Long Short Term Memory, generating symbol-embeddings from a natural language sentence, Meanwhile, we may have the encoder function, implemented as a classical Transformer, generating the context-embedding from the remaining document.
>
> We will revise the manuscript to make these points clear.
>
> ### Weakness 3
>
> > Weaker semantic relation might broaden the applicability.
>
> We agree. We speculate that, using a weaker relation (namely semantic similarity) would lead to similar but more general results. Formally, we could define semantic similarity as follows:
>
> $$u \circeq_\epsilon v \iff d(p(Y|u,\mathrm{P}),p(Y|v,\mathrm{P})\leq \epsilon$$
>
> Where $d$ is a distance function (e.g. Euclidean or Weisserstein). We anticipate that two semantically similar symbols must be encoded in close embedding by the optimal embedding function.
>
> > Weaker alignment might broaden the applicability.
>
> We agree. Again, we speculate that using a weaker hypothesis would lead to similar but more general results. A weaker definition of alignment could be made as follows:
>
> $$d(p_{\mathcal{D}}(y|u,\rho),p_{\mathcal{D}}(y|v,\rho)) \leq \epsilon \iff d(p_{\mathcal{C}}(\rho|u),p_{\mathcal{C}}(\rho|v)) \leq \epsilon $$
>
> Here, $d$ is a distance function (e.g. Euclidean or Weisserstein). We speculate that when this weaker hypothesis hold, one could derive a weaker version of the Corollary 4.3:
>
> $$u \circeq_{\epsilon_0} v \iff d(\mathcal{E}^*(u),\mathcal{E}^*(v)) \leq \epsilon_1 $$
>
> Here, $d$ is a distance function (e.g. Euclidean).
> We will include this consideration as potential avenues for future work in the revised manuscript.
>
> [1] Maurer et al. The Benefit of Multitask Representation Learning. 2016.

---

> > ### Author Response · Authors · 2024-11-16
> > **Change Log**
> >
> > We have updated the manuscript to include modifications requested by the reviewers. Below is a log of the changes made. We hope these revisions have enhanced the quality and clarity of the manuscript.
> >
> > ### Main Changes
> > - Added a backward proof for Corollary 4.3 in Appendix Section A.4.
> > - Clarified the assumption $p(\rho|u) = p(\rho|v)$ in the "Threats to Validity" section (Section 7).
> > - Addressed assumption $p(\rho|u) = p(\rho|v)$ in Section 3 and 4.
> > - Addressed the issue of embedding distances not being zero (Section 5).
> > - Addressed alignment hypothesis in the "Threats to Validity" section (Section 7).
> > - Provided a stronger motivation for the basis hypothesis (Section 3).
> > - Corrected minor typographical errors.
> > - Added a Word2Vec-style experiment in Appendix Section A.11.
> > - Improved terminology by revising the use of "semantic relations" (often in the context of "semantic equivalence relations") and "semantic learning" to avoid misinterpretation related to linguistic semantics.
> > - Revised the abstract, introduction, discussion, conclusion, and threats to validity sections to better clarify the scope and contributions of the work.
> > - Specified "semantic relations" in RQ2.
> > - Provided a more precise answer to RQ2.
> > - Added a "Future Work" section (Section 10), which includes:
> >   - Exploring multitask learning parallelism.
> >   - Considering the weakening of assumptions.
> > - Included a cat-vs-dog example in Section 2.1.
> >
> > ### Added References
> >
> > - Tongzhou Wang and Phillip Isola. *Understanding contrastive representation learning through alignment and uniformity on the hypersphere.* In *International Conference on Machine Learning*, pp. 9929–9939, PMLR, 2020.
> > - Levy, Omer, and Yoav Goldberg. *Neural word embedding as implicit matrix factorization.* Advances in Neural Information Processing Systems 27 (2014).
> > - Andreas Maurer, Massimiliano Pontil, and Bernardino Romera-Paredes. *The benefit of multitask representation learning.* Journal of Machine Learning Research, 17(81):1–32, 2016.
> >
> > ### Awaiting response on
> > - Experiment with alternative contrastive architecure.
> > - Experiment with triplet loss instead of InfoNCE loss.

---

> > ### Comment · Reviewer_iHA7 · 2024-11-28
> > **feedback**
> >
> > Thank you for the detailed feedback and the revisions. While I appreciate the rigorous approach you pursued, I still find the setup and assumptions overly simplistic to have significant practical relevance/ impact. Nevertheless, I maintain my score and still in favor of accepting the work, as it careflly articulates some of the intuitions regarding the connection between embeddings and (the specific form of) semantic equivalence.

---

### Official Review · Reviewer_fYp9 · 2024-11-08

**Soundness:** 3
**Presentation:** 3
**Contribution:** 3
**Rating:** 5
**Confidence:** 2

**Summary:**

This paper explores the theoretical foundation of contrastive learning, a popular self-supervised technique used to generate high-quality embedding representations across various data modalities (e.g., images, audio, text). While contrastive learning has shown empirical success in encoding semantically similar objects into close embedding representations, a formal understanding of this process is lacking. To address this gap, the authors propose a formalization of semantic equivalence in contrastive learning, inspired by principles from programming language theory. They introduce the distributional alignment hypothesis, which posits that the alignment of distributions in contrastive tasks is essential for effective downstream performance. Through analysis of the SimCLR method, they demonstrate that contrastive learning can inherently encode semantically equivalent symbols in close proximity within the embedding space.

**Strengths:**

1.The paper provides a theoretical perspective on contrastive learning, bridging the gap between empirical success and formal understanding.

2.Introducing the concept of semantic equivalence in the context of contrastive learning is innovative, borrowing ideas from programming languages to define how two symbols can be considered equivalent in the embedding space.

3.The proposal of the distributional alignment hypothesis offers a new framework for understanding when contrastive tasks are effective for downstream applications, potentially guiding future work in contrastive learning model design.

**Weaknesses:**

1.While the theoretical findings are compelling, the paper might benefit from empirical experiments to validate the proposed hypotheses, particularly the distributional alignment hypothesis, across different contrastive learning frameworks. As the study is heavily focused on theoretical formalism, which may limit its immediate applicability for practitioners who are looking to implement contrastive learning solutions without deep theoretical knowledge.

2.The paper’s formalism of semantic equivalence is based on an analogy to programming languages, which may not translate perfectly to the nuances of different data modalities (e.g., images vs. text), potentially limiting its generalizability across tasks.

3.There have been a number of analysis to study the effectiveness of contrastive learning, but it is hard to say how the new perspective would help the improvement of contrastive learning method.

**Questions:**

Please refer to the weaknesses

---

> ### Comment · Reviewer_fYp9 · 2024-12-03
> **Waiting for Responses**
>
> Actually, I think the theoretical findings in this paper are convincing, but I really want to see more empirical results to test its generalization ability. By the way, there have been massive theoretical work for studying contrastive learning, their major difference with this work is also not clear.
> Therefore, it is hard for me to know how valuable this work is, and whether it is worth to be accepted by this conference.

---

> > ### Author Response · Authors · 2024-12-03
> >
> > We thank the reviewer for their thoughtful feedback and encouraging comments. Below, we address their concerns.
> >
> > ---
> >
> > > Actually, I think the theoretical findings in this paper are convincing, but I really want to see more empirical results to test its generalization ability.
> >
> > We appreciate the positive remarks about our theoretical findings. While we are unable to add new experiments at this stage, we have included an additional word-embedding-inspired experiment in Section A.11 of the appendix, following Reviewer 5kK9's suggestion. This experiment evaluates semantic distances between three categories of words:
> >
> > 1. **Artificially constructed synonyms**: When replacing a word $ u $ with an artificial synonym $ v $ such that $ p(u|\rho) = p(v|\rho) $, their embeddings become extremely close, consistent with our theory.
> > 2. **Naturally occurring synonyms**: For 20 words, we tracked the distances to 20 semantically similar counterparts during training. These embeddings also became close, though not as tightly as the artificial synonyms.
> > 3. **Different words**: For the 100 most common words with no synonymity relation, embeddings were farther apart, aligning with our theoretical predictions.
> >
> > Previously, we proposed ablations with Barlow Twin architecture and TripleLoss, but we did not receive feedback on these suggestions.
> >
> > Our work aims to theoretically ground the observed phenomenon in contrastive learning: embeddings naturally organize into semantically relevant structures. For example, images of two dogs are closer than those of a dog and a clear sky. Consequently, we prioritized theoretical analysis over additional empirical validation.
> >
> > ---
> >
> > > There have been massive theoretical works for studying contrastive learning; their major differences with this work are also not clear.
> >
> > We acknowledge this concern. Unfortunately, we cannot modify the manuscript at this point. However, we clarify how our work differs from existing literature:
> >
> > 1. **Related Works**:
> >    - [1] and [2] are the two most closely related works.
> >    - Both focus on the connection between contrastive learning and downstream tasks, assuming a linear classifier.
> >    - [1] assumes the same distribution for contrastive and downstream data and the existence of a latent label distribution.
> >    - [2] uses a latent variable representing downstream data, conditionally independent of the data given the label.
> >
> > 2. **Key Differences in Our Framework**:
> >    - We derive semantic equivalence from programming language theory rather than a latent distribution.
> >    - Our notion of "good representation" emphasizes that semantically similar symbols are encoded as close vectors, distinct from assuming linear separability.
> >
> > We hope this clarifies the novelty and theoretical contributions of our work.
> >
> > ---
> >
> > References:
> > [1] Sanjeev Arora et al., *A theoretical analysis of contrastive unsupervised representation learning*, ICML 2019.
> > [2] Jason D. Lee et al., *Predicting what you already know helps: Provable self-supervised learning*, NeurIPS 2021.

---

### Author Response · Authors · 2024-12-03
**Discussion Period Summary**

To assist reviewers, area chairs, and senior area chairs, we provide a summary of the key points raised in the discussion.

---

### Limited Empirical Evidence

A common concern among reviewers was the lack of convincing empirical evidence supporting the theoretical findings.

To address this concern, and following Reviewer 5kK9's suggestion, we introduced a word-embedding-inspired experiment to provide empirical validation in a more realistic setting. While we proposed additional experiments, they were not included as the reviewers did not provide feedback on them.

---

### Concerns About Theoretical Assumptions

Reviewers expressed concerns about the strength of assumptions:
- $ p(u|\rho) = p(v|\rho) $,
- $ p(y|u,\rho) = p(y|v,\rho) $ and,
- the specific contrastive learning framework.

We revised the manuscript to clarify the theoretical limitations deriving from these assumptions. We also emphasized their necessity, given the absence of a label distribution in  the contrastive data. Nonetheless, these assumptions lead easily to the alignment hypothesis, which we believe is a valuable theoretical insight.

We note in the future work section that relaxing these assumptions may lead to a weak version of the alignment hypothesis.

---

### Concerns About Terminology

Reviewer t6m4 noted that the term *semantics* has a strong linguistic connotation that differs from our formalization, potentially causing confusion.

In response, we revised the manuscript to clarify our use of terminology and ensure consistency with the proposed framework.

---

### Limited Practical Implications

Reviewers raised concerns about the limited practical implications of our work.

Our aim is to provide a theoretical understanding of contrastive learning’s inner workings, specifically how contrastive embeddings organize semantically. The main practical implication lies in offering a new perspective rather than direct applications. We revised the manuscript to better articulate these implications, as well as the work’s limitations and potential future directions.

---

### Strength of Results

Several reviewers questioned the alignment hypothesis (AH), suggesting it may be too strong to hold in practice.

We agree that the AH is a strong assumption and difficult to verify empirically, a point now emphasized in the manuscript. Nevertheless, we argue that it offers valuable theoretical insights into why contrastive pretraining is effective for specific contrastive-downstream data pairs.

---

### Missing Proof

Reviewer 5kK9 noted the absence of a backward proof for Corollary 4.3.

This missing proof has been added to the manuscript.

---

### Concerns About Related Literature

Reviewer fYp9 found it difficult to discern the novelty from between our work and prior literature.

Although we could not revise the manuscript during the discussion period, we provided a detailed comparison with the two most related works ([link](https://openreview.net/forum?id=6EadiKkfgR&noteId=ol56KABS49)). These comparisons will be included in the final version.

---

### Additional Related Works

Reviewer iHA7 suggested a connection between our work and multi-task learning literature.

We expanded on this insight in the future works section to outline possible connections and implications.

---

### Meta-Review · Area_Chair_gy7i · 2024-12-24

**Metareview:**

This paper provides a formal treatment to understanding the learning of contrastive representations, by considering the idea of semantic equivalence and proposing a distributional alignment hypothesis. The key idea explored in this paper is that two tasks are distributionally aligned if they share semantic equivalence; “that is, if tokens are semantically equivalent in one task, they remain so in another”. The paper presents an analysis with the SimCLR method.

**Strengths:** The paper presents an interesting theoretical understanding of contrastive learning. Reviewers found the theoretical findings compelling, clearly written, and to offer novel insights.

**Weaknesses:** The work could benefit from more rigorous empirical experiments to support the main theoretical arguments of the work. Reviewers found the assumptions overly simplistic, therefore limiting the impact or practicality of the proposed method.

**Reason for rejection**: The work, though lacking empirically, does clearly articulate some of the intuitions underlying contrastive learning and semantic equivalence, albeit under some simplistic assumptions. It can be a starting point for further research on theoretical understanding of similar methods. At this time, due to the limitations pointed out by reviewers (extra strong assumptions, limited practical applicability), perhaps the work is not ready for publication.

**Additional Comments On Reviewer Discussion:**

Based on the reviewer feedback, the authors have provided changes in the paper that clarify some concepts and describe methods clearly. The authors however do not add additional experiments due to a lack of engagement / enthusiasm by the reviewers. However, the authors do provide many additional changes in the paper as a result of the reviewer feedback, which helps strengthen it for another version of the work.

---

### Decision · Program_Chairs · 2025-01-22

Reject